# Characteristics of marine shipping emissions at berth: profiles for PM and VOCs

Qian Xiao[1,2#], Mei Li[3#], Huan Liu[1,2*], Fanyuan Deng[1,2], Mingliang Fu[1,2], Hanyang Man[1,2], Xinxin Jin[1,2], Shuai Liu[1,2], Zhaofeng Lv[1,2], Kebin He[1,2]

[1] State Key Joint Laboratory of ESPC, School of Environment, Tsinghua University, Beijing 100084, China

[2] State Environmental Protection Key Laboratory of Sources and Control of Air Pollution Complex, Beijing, 100084, China

[3] Atmospheric Environment Institute of Safety and Pollution Control, Jinan University, Guangdong, 510632, China

*Correspondence to*: Huan Liu (liu_env@tsinghua.edu.cn)

[#]These authors contributed equally to this work.

**Abstract.** Emissions from ships at berth play an important role in the exposure of atmospheric pollutants to high density population in port areas, but these emissions have not been understood very well. In this study, volatile organic compounds (VOCs) and particle emissions from 20 container ships at berth were sampled and analyzed during the fuel switch period at Jingtang Port in Hebei Province, China. VOCs and particles were analyzed by gas chromatography-mass spectrometer (GC-MS) and the Single Particle Aerosol Mass Spectrometer (SPAMS), respectively. VOCs analysis showed that alkanes and aromatics, especially benzene, toluene and heavier compounds eg., n-heptane, n-octane, n-nonane, dominated the total identified species. Secondary organic aerosol yields and ozone forming potential were $0.017 \pm 0.007$g SOA/g VOCs and $2.63 \pm 0.37$g $O_3$/g VOCs, respectively. Both positive and negative ion mass spectra from individual ship were derived and intensity of specific ions could be quantified. Results showed that element carbon (35.74%), element carbon-organic carbon mixture (33.95%) and Na-rich particles (21.12%) were major classes with a total number fraction of 90.7%. Particles from ship auxiliary engines were in a size range of 0.2 to 2.5μm, with a peak occurring at around 0.4μm. The issue of vanadium as tracer element was discussed that V was not a proper tracer when using low sulfur content diesel oil. The average percentage of sulfate particles from shipping emissions before and after switching to marine diesel oil kept unchanged at a level of 24%. Under certain wind direction with berths on upwind directions, the ratio before and after January 1st is 35% and 27% respectively. The total results provide

robust evidences in port area air quality assessment and source apportionment.

## 1. Introduction

Emissions of multiple pollutants and greenhouse gases like $NO_x$, $SO_2$, particulate matters(PM) and $CO_2$ from ocean-going ships have been significant sources to global air pollution(Eyring et al., 2010;Corbett et al., 1999), of which the impacts were on a global scale and widely-spread. Along with the impacts on air quality(Vutukuru and Dabdub, 2008) and climate change, health impacts were also revealed and researched by plenty of recent studies(Andersson et al., 2009;Corbett et al., 2007). There were about 60,000 cases of deaths due to cancer and cardiovascular diseases associated with ship exhausts worldwide(Tian et al., 2013). In East Asia, where 8 out of 10 top container ports were located, ship emissions caused premature deaths of between 14,500 to 37,500 in 2013, almost doubled comparing to that in 2005(Liu et al., 2016).

Emission characteristics, including size distribution of particles, chemical compositions of particles and volatile organic compounds (VOCs), are with particular importance to understand the climate and health impacts of shipping. Plenty of studies on emission factors and chemical characteristics were conducted throughout China and western countries. In terms of domestic gaseous pollutants and PM emission factors, CO, HC, $NO_x$ and PM emission factors during maneuvering and cruising conditions from 7 in-use ships (4 container ships included) were tested by portable emission measurement system, illustrating the nucleation and accumulation mode of major PM (Peng et al., 2016), and in another case 3 offshore vessels were tested by on-board systems(Zhang et al., 2016). Black carbon in exhausts from 71 ships were analyzed in California(Buffaloe et al., 2013). Apart from those direct sampling of shipping exhausts, plumes were sampled via aircraft 100km off the California coast(Chen, 2005) and in-port ambient PM measurements were also done in the Port of Dock using aerosol time-of-flight mass spectrometer(Healy et al., 2009). Chemical properties of PM in plumes of 1100 commercial ships were studied by time-of-flight aerosol mass spectrometer in Mexico Gulf(Lack et al., 2009). In summary, these listed studies generally focused on emission factors of typical gaseous pollutants and PM, and plume studies were more relevant to the mixture of primary and secondary aerosol, result in insufficiency of directly-exhausted gas and PM studies. Another noticeable fact was that these studies on emission characteristics mostly explored the at-sea parts of shipping emissions, thus paying no adequate attention on specific at-

berth conditions.

Based on the studies mentioned above, it could concluded that one of the major gaps in exploring emission characteristics is to understand specific shipping emissions under at-berth mode. The significance of at-berth emissions was attributed to the fact that contribution of emissions under operation mode of 'at berth', mainly dominated by the auxiliary engine emissions, increased rapidly as the target areas shrank from global scale to the port regions. For global shipping emissions, the share of emissions from auxiliary engines under at-berth mode was between 2-6%(Corbett and Koehler, 2003). When the domain was limited to a region-scale, eg. East Asia, the contribution of $SO_2$ and $NO_x$ emissions at berth increased to 26%(Liu et al., 2016). When focusing on the national costal area, the shares continued growing to 28% and up(Fu et al., 2017). An in-port shipping emission inventory of Yangshan Port in China(Song, 2014) revealed that auxiliary engine exhausts contributed 40.5% and 43.3% to overall $PM_{2.5}$ and $SO_x$ respectively. This is why the Fuel Switch at Berth regulation was proposed in 2011 and enforced in July 2015 in Hong Kong, then proposed in 2015 and enforced in 2017 in Main land China as the first step on marine fuel quality control. Yet no valid and comprehensive studies on chemical properties of auxiliary engine exhausts have been conducted in China, which owned 12 of 20 listed world's largest ports by 2010(Report, 2013). Consequently, the lack of studies on chemical characteristics of at-berth emissions became a barrier to further analyze the mechanism of organic aerosol transition and toxicology of human health. Owing to the lack of these specific studies, it was more difficult to apply accurate at-berth data in evaluating health impact instead of using general average of shipping emission data without distinguishing both at-sea emission and at-berth emission.

Apart from the significance of at-berth emissions, another critical issue was the impact of different fuels on shipping emission chemical characteristics. The main fuel types applied in ship main engines or auxiliary engines were mainly residential oil, heavy fuel or intermediate fuel oils (Hays et al., 2008) as well as marine diesel oil. Low-grade heavy fuel oil (HFO), known as bunker oil or residual oil, which usually had sulfur content of higher than 0.5% and metallic elements such as vanadium, nickel and copper, was commonly used in marine engines and was responsible for high level of PM and gaseous pollutants such as $SO_X$(Agrawal et al., 2009). In contrast, marine diesel oil (MDO) was lighter and cleaner diesel with lower sulfur and metallic element content(Corbett and Winebrake, 2008). Aiming at reducing the emissions, More stringent limit on fuel sulfur content and switching to cleaner marine diesel fuels became a common trend(IMO, 2017). Various studies illustrated the distinction of using HFO and MDO in

shipping emission chemical characteristics. For black carbon, using low sulfur content MDO could result in a reduction of up to 80% of total BC emissions comparing to using HFO(Lack and Corbett, 2012). Recent comparative study focused on HFO and MDO in Europe sulfate emission control area (SECA) and results showed that a decline of sulfur content from 0.48% to 0.092% lead to a reduction of 67% in

PM mass and 80% in $SO_2$ emission(Zetterdahl et al., 2016). Another comprehensive study including PM, EC, heavy metals was made to make comparison between HFO and standardized diesel fuel (Streibel et al., 2017). Following the regulations of using low sulfur fuels implemented by western countries, China began to launch its own stepwise regulations associated with domestic emission control areas (DECAs) to minimize conventional pollutants from shipping emissions. Under the new regulation ships were

required to use diesel with a lower sulfur content of below 0.5% at berth, corresponding to the vital role that at-berth emissions played in air quality. Fuel switch would lead to changes in chemical characteristics of ship-exhausted gas-phase VOCs and PM, which were closely relevant to ambient air quality in port areas and health impact on population. Nevertheless, previous comparative studies on different fuels tended to emphasize the diversity in total amount and emission factors, there lacked of studies revealing

chemical characteristics caused by fuel switch especially in China.

In order to explore the chemical composition of VOCs and particulate matters (PM) from ship auxiliary engines, this research was located in a key port area and was designed to cover the primary period of new policy implementation: from December 27th, 2016 to January 15th, 2017 in Jingtang Port, which was among the pilot key ports where new regulation came into effect since January 1st, 2017. An amount of

20 container ships were sampled and measured for VOC and PM emissions from auxiliary engines in the mode of at-berth. Via the application of gas chromatography-mass spectrometer (GC-MS) and single particle aerosol mass spectrometer (SPAMS), this research conducted a comprehensive exploration in perspectives of VOC profiles, PM size distribution and typical ion mass spectra, which could act as fundamental to impact assessment of shipping emissions and source apportionment in key port regions.

**2    Experimental and methods**

**2.1   Sampling methods and instruments**

**2.1.1   Information of Jingtang Port, sampling sites and ships**

Ambient sampling site was located inside Jingtang Port, Tangshan City, Hebei Province, China. Jingtang

Port is located in Bohai Bay and belongs to Port of Tangshan, which is among the core ports in domestic emission control area. According to the China Port Yearbook 2015, the annual traffic of ships in Port of Tangshan reached 15084 and the total throughput exceeded 500 million tons, ranking 5th among global port throughputs. Jingtang Port area is surrounded by the Port Economic Development Area, which has

a population of 78, 300. Tangshan is a typical industrial city with average $PM_{2.5}$ concentration in winter of 117 μg/m$^3$ (Zhang et al., 2017). The current $PM_{2.5}$ source apportionment studies in Tangshan did not include shipping emissions due to the lack of basic information and researches. The background information indicates the significance and urgency of studying the impact of shipping emissions and the effect of the fuel switching policy.

As is shown in **Fig. 1(a)**, the center of population mainly concentrates in the residential area, located in the north of the port area, about 2km away from the port. About 2.5km in the west of the port area there is a thermal power plant with after-treatment facilities according to the strict emission control standards to power plants in China. Between the port and the other zones are two main roads with trucks driving to carry containers in and out of the port, which is about 1km away from the sampling site. Besides trucks

and the power plants, there's no further emission sources near the port area.

The site for ambient particle collection and instrumental analysis is surrounded by the four pools and the channel, located on an open and flat corner close to the #26 and #27 Berth as well as the container yard inside the port, as is shown in **Fig. 1(b)**. No tall buildings exist around the sampling instrument. The distribution of berths, pools and the sampling site guarantees that plumes from ships at berth are prone

to reach the sampling instrument.

The information of 20 container ships included in this study was collected via on-board inquiry and was listed in **Table 2**.

**2.1.2  VOCs sampling and analysis by GC-MS**

VOCs from ship at berth were sampled via Entech summa canisters with standard volume of 3.2L. When

auxiliary engines were in operation, Teflon tubing with a length of 1m was stretched into the exhaust pipe, with the other end linked to summa canister. The flow rate was kept constant by Entech CS1200ES passive canister sampler, which also filtered impurities like particles and ashes. The first patch of samples was regarded as preliminary tests in order to determine proper dilution factors and sampling time and

flow rate. Sample dilution was conducted by Entech 4600 dynamic diluter with various dilution factor range from 10 to 80, depending on the original sample concentration. Agilent 5975C-7890A was calibrated with standard gas and used for analyzing diluted VOC samples. A total of 93 VOC species were detected and mass percentage of single compound could be calculated according to sample inlet volume, dilution factor and calibration curve. Owing to the limit of chromatographic column property equipped on the GC-MS, alkanes and olefins with carbon number smaller than 4 were not detected. VOCs with carbon number larger than 6 are more relevant to the yield of secondary organic aerosols (Gentner et al., 2012).

### 2.1.3    Particles sampling and analysis by SPAMS

Ship exhaust particles were collected directly from exhaust pipes of auxiliary engines on ships by Tedlar bags and metal tubing designed specifically for particle sampling. The whole sampling process was achieved by a non-contact sampling box and air pump. Then samples were sent to the Single Particle Aerosol Mass Spectrometer from Hexin Analytical CO., Ltd(Li et al., 2011)as soon as possible to be analyzed. SPAMS shared common principle and mechanism with Aerosol Time-of-Flight Mass Spectrometer (ATOFMS). It was frequently applied in online measurement and analysis of single particle aerosol from heavy diesel vehicle exhausts(Shields et al., 2007) and biomass burning(Bi et al., 2011;Xu et al., 2017).

Ambient particle sampling was conducted from December 27th, 2016 to January 15th, 2017, spanning about 20 days. Ambient particles were sampled and analyzed by SPAMS, with the inlet fixed at a height of 3.6m from the ground level.

### 2.2    Ion mass spectra identification and manual grouping

Ion mass spectra with positive and negative ion information were derived from SPAMS output as fundamental for further analysis. Both positive and negative ions were in an m/z range of 1-250. Based on ART-2a neural network algorithm(Song et al., 1999), single particles sharing similar mass spectral signatures were further grouped into clusters. The whole algorithm process was operated on MATLAB 2011a, and a vigilance factor of 0.7 as well as a learning rate of 0.05 was set respectively with 20 iterations. Then further manual classification was demanded in order to merging particle clusters into expected types, depending on the target of research. Clusters that counted for top 95% of all particles would be

analyzed according to the method mentioned previously.

To identify each type of particles, ions with certain m/z value were used as markers(Fu et al., 2014). For element carbon (EC) particles, [$C_n$] signals were the most typical markers distributing in both positive and negative ion mass spectra. For organic carbon (OC) particles, ions with m/z=29, 37, 41, 43, 51, 61, 63,...represent $C_2H_3^+$, $C_3H^+$, $C_3H_5^+$, $C_4H_3^+$, ..., respectively. Meanwhile, there existed a status of EC ions signals mixed with typical OC signals, which should be classified into EC-OC mixed particles (ECOC), and it was hard to accurately divide them into EC or OC types owing to the coexistence of EC and OC ion signals in single particle spectrum. ECOC particles were formed by EC or VOCs that had been oxidized in the air into OC attaching on the surface of the EC particles(Liu et al., 2003). $Na^+$ came both from sea salt and contents existed in fuels, especially in heavy fuel oil or residual oil. $K^+$ came from biomass burning, sea spray sources as well as the fuels(Leeuw et al., 2011). $V^+$ and its oxidized ion $VO^+$ were considered as the symbol of ship exhaust for V existed in fuel like heavy fuel oils (HFO). Previous studies (Celo et al., 2015;Liu et al., 2017) frequently used $V^+$ and $VO^+$ to identify particles from ship exhausts, though they could be observed more rarely comparing to other large-quantity particles. Other metallic ions such as $Fe^+$ and $Cu^+$ could also be observed, and they had relatively small relative intensity, for the metal elements scarcely existed in atmosphere and fuels combusted in powertrain systems. In most cases, particles with relative intensity of metallic ions higher than 0.05 and no other obvious high signals could be classified into metal rich particles.

## 3    Result and Discussion

### 3.1    VOCs speciation

Exhausts from a total of 20 ships have been sampled on board and all samples have been diluted to a concentration of approximately 3-4 ppm in order to guarantee the validity and accuracy of GC-MS analysis. 93 species were detected for each ship sample. The sum of mass concentration of identified 93 species was defined as 1, thus normalizing the mass concentration of single species. 4 samples were excluded for their irregularity after 3-sigma test for all data. Then the remaining 16 samples were averaged by percentage of mass concentration and then VOCs speciation profile was obtained for all 16 ships, which shared similarity in species mass concentration distribution. The histogram of VOC profile by mass percentage was shown in **Fig. 2** and top 32 species were listed in **Table 3**. Alkanes and aromatics

dominated the total identified VOCs from ship auxiliary engine exhaust. N-heptane, methylcyclohexane, n-octane, n-nonane, n-decane and n-undecane contributed considerably to alkane emissions, which indicated that alkanes with carbon number of more than 7 were more likely to be emitted from ship diesel engines compared with other mobile sources. Among aromatics benzene and toluene contributed approximately 9% to total VOCs emissions. This result was acceptably in consistence with study of Huang et al. on diesel emissions in Shanghai(Huang et al., 2015). In view of oxygenated VOCs and haloalkane contents, acetone and $CH_2Cl_2$ exceeded benzene and toluene. However, there were considerable standard deviations in average oxygenated VOCs and haloalkanes mass proportion due to obvious ship-based difference and quantification inaccuracy of GC-MS for these two classes of compounds.

### 3.2 SOA Yields and OFP by VOCs from ship exhausts

Based on VOCs profile, SOA yields and ozone forming potential for ships were calculated. SOA yield value for individual precursor and definition of non-precursors were referenced from Gentner's study(Gentner et al., 2012). Here was a noticeable fact that intermediate VOCs (IVOCs) were not identified and quantified in this study, which may cause underestimation of the actual SOA yields. Also, VOC source profile of 3 types of diesel trucks (light-, middle- and heavy-duty truck respectively) (Yao et al., 2015) and profiles of heavy-duty diesel trucks in Huang's study (Huang et al., 2015) were referenced to calculate and make comparison. Average profile for light-duty passenger gasoline vehicles were collected from previous research in China (Cao et al., 2015) to make comparison among them. Considering this comparison could provide insights to emission control strategy and fuel quality for different sectors in the same country, the comparison made among studies in China would make more sense. The results were presented in **Fig. 3**, and the average SOA yield for 16 ships measured was 0.017g SOA/g VOCs. As seen in **Fig. 3**, conclusion can be drawn that in terms of SOA yields, ship-exhaust VOCs would generate more SOA than that of diesel trucks as well as gasoline vehicles. The main reason was the content difference of heavy organic compounds. Ship exhausts contained more heavy VOCs like alkanes and aromatics with carbon number of more than 9 than that of vehicles, among which aromatic contents like benzene, toluene and xylene were especially responsible for SOA yields. The unconsidered IVOCs tended to exist more commonly in heavy fuels rather than diesel oil or gasoline.

Ozone forming potential showed different trend against SOA generation in that VOCs from ship exhausts

had approximately equal OFP with diesel and gasoline vehicles. According to the calculating method of maximum increment reactivity (MIR)(Carter, 1994), lighter VOC had higher MIR scale, which meant more contribution to ozone formation. VOCs from ship exhausts had relatively lower content of light hydrocarbons, thus lowering the overall ozone forming potential. The average OFP of ship emitted VOCs was 2.63g $O_3$/g VOCs. In conclusion, VOCs from ship exhausts might play a more important role in secondary organic aerosol formation in port area than diesel trucks and gasoline vehicles.

### 3.3 Overall particle characteristics by SPAMS

### 3.3.1 Average ion mass spectra

Samples collected by Tedlar bags and glass bottles from 20 ships at berth were analyzed by SPAMS. Excluding some invalid samples with very few particles, the average ion mass spectra of both positive and negative ions expressed in relative area were obtained after SPAMS analysis and classification, as shown in **Fig. 4**.

The average ion mass spectra presented overall characteristics of total 20 ships. For positive ion spectrum, it could be noticed that $Na^+$ signal was observed as the highest peak. Another sort of abundant ions represented EC and OC, occurring in the form of $C^+$, $C_2^+$, $C_3^+$… as EC and $C_3H^+$, $C_2H_3^+$…as OC, respectively. It was easy to find that the relative signal area of EC ions were higher and more widely distributed than OC ions, which was consistent with previous researches that fuel combustion produced more EC than OC particles. Though not as obvious as those abundant sorts above, the percentage of $V^+$ and $VO^+$ were relatively low but not negligible, which were commonly used to identify ship source particles.

For negative ion spectrum, the components were not as complex as positive. The highest peak was $HSO_4^-$ with m/z=-97, which exceeded 50% of the overall negative ion relative area. $HSO_4^-$ was typical marker of sulfate particles, which were formed via secondary reaction in atmosphere. This part will be further discussed in following sections. EC signals such as $C_2^-$, $C_3^-$, and $C_4^-$ could also be observed in negative spectrum. In addition, markers of nitrates with m/z=46 and 62 were $NO_2^-$ and $NO_3^-$. Nitrates were regarded as secondary particles that might come from the aging of primary particles emitted directly by fuel combustion or the transformation of gas-phase $NO_2$. The relative area of nitrate signals was much lower than $HSO_4^-$, illustrating that nitrate particles were not predominant in ship exhausts and not the

main research goal in this study.

### 3.3.2 Manual grouping results

Based on single particle ion spectrum similarity and ART-2a neural network algorithm, particles from ship exhaust were further grouped into clusters of 175, among which the top 86 clusters covered 95%, with the rest defined as others. Clusters were set as OC, EC, ECOC, Na (Na-Rich), K (K-rich), V (V-rich), Cu (Cu-rich), Fe (Fe-rich), Mn (Mn-rich) and others, depending on the similarity and manual grouping. Method and criterion of the manual grouping mainly focused on positive ions. Particle grouping could be a fundamental access to accurate source apportionment of a target region.

After that a composition analysis was conducted and result was shown in **Fig. 5**. It was shown that EC and ECOC particles dominated with a proportion of 35.74% and 33.95%, while Na-rich particles ranked 3rd with percentage of 21.12%. Data of EC and ECOC indicated that carbon emitted by fuel combustion tended to product more EC and ECOC than purely OC particles. The amount and fraction of OC particles was remarkably lower than EC and ECOC with a factor of ~10. Meanwhile, OC tended to attach to EC particles and thus forming ECOC particles and increasing the fraction of ECOC particles. In view of the composition of fuel combusted by marine engine, Na does exist with a mass concentration of around 13-22mg/kg in several samples of heavy fuel oil (HFO) and lower content in marine diesel oil(Moldanová et al., 2013;Lack et al., 2009). Moldanova (Moldanová et al., 2009) analyzed particle from ship exhausts by using two-step laser mass spectrometry and $Na^+$ was observed with obvious peak in the mass spectra. According to Sippula's (Sippula et al., 2014) research on marine diesel engine, Na can also exist in lubrication oil and be exhausted along with the lubrication and grinding process. It is widely recognized that the metal content existed in exhausted particles has strong correlation with that in fuel (Celo et al., 2015;Moldanová et al., 2013). Although the content of Na is obviously lower than that of vanadium (55-133.8mg/kg), the peaks of $Na^+$ in ion mass spectra are mostly higher than peaks of $V^+/VO^+$ signal. This phenomenon indicate that there are external sources of Na and one of the most possible sources is sea salt, whose typical elements are Na. In conclusion, the Na-rich particles come from a multiple source of both fuel composition and sea salt perturbation. Only particles with relative area of $Na^+$ obviously higher than other positive ions were counted as Na-rich particles. The same rule also adapted to K-rich particles. Owing to the fact that the content of K in most marine fuels is lower than that of Na, V and other typical metallic elements, one of the possible sources of K-rich particles is sea salt (O'Dowd et al., 1997), which

is similar to the source of Na-rich particles. $Na^+$ and $K^+$ were both widely spread among all particles. V-rich particles were likely to contain mixtures of ammonia sulfate or sulfuric acids(Divita et al., 1996), which could be proved by ion mass spectra. Particles with relatively high $V^+/VO^+$ signals occupied only 1.1% according to the classification criterion, which were contributed by a few numbers of ships. This will be discussed in following parts.

### 3.3.3 Size distribution

Particles with aerodynamic diameter ranging from 0.2μm to 2.5μm, covering the accumulation mode (0.1μm to 1μm) and coarse mode (over 1μm)(Frick and Hoppel, 2010), were analyzed by SPAMS and this size range has a significant correlation with $PM_{2.5}$ pollution, which should be evaluated seriously. Size information of 257911 particles from 20 ships were derived from SPAMS and 19342 of them were hit. It showed a skewed distribution with a peak appeared between 0.38μm and 0.44μm. This result disagreed with the measurements conducted in Ireland using SMPS(Healy et al., 2009), in which particles measured were mostly within the size range of 100nm. Moreover, bimodal distribution was not observed for the reason of limited size range measurement of smaller than 2.5μm, while the coarse mode usually occurred beyond that. The divergence might result from the difference of measuring instruments and fuels. 95% of all particles with size information lay in 0.2μm-1.46μm, which implied that particles from ship auxiliary engine exhaust could be classified into fine particles, contributing to the $PM_{2.5}$ in ambient air. It is remarkable that the magnitude of size distribution in this study was consistent with some of the test results by engines using different fuels. Most of the samples in this study were collected after January 1[st], when new policy of forcing ships to switch to clean diesel fuel had taken effect. This could result in the decline of particle aerodynamic diameter.

In order to further analyze the size distribution information of different sorts of grouped particles, **Fig. 6** was drawn to reveal the proportion variation among different particles over size range. As OC particles only occupied less than 5% of overall particles, they show no apparent trends and distributed dispersedly in each section. It can be observed that in sections of larger than 1.30μm OC particles occasionally "disappeared" in some bins, while they kept continuously between 0.2-1.30μm, which might indicated that OC tended to concentrate in a relatively tiny size section. The rule of EC particle distribution was a little similar to bimodal distribution, with the first peak settled in the section of 0-0.6μm. In the section of 1.75μm to 2.5μm, it appeared to present another peak but not as obvious as the first one. Accordingly,

in section of 0.2-0.5μm, proportion of ECOC particles was lower than that in the section of over 0.5μm, implying that ECOC tended to form larger particles comparing to EC. Another obvious fact was that K-rich particles were more likely to present normal distribution in the range of 0.25μm to 1.0μm, which was discrepant with size distribution of Na-rich particles. Na-rich particles distributed almost equally among bins in 0.2-1.25μm, and proportion increased starting from 1.25μm. Metal contained particles were observed between 0.25μm to 1.25μm, occupying much lower proportion comparing to other massive sorts of particles. Metal contained in particles emitted by ship engine mainly came from combustion of heavy fuel oil and were known to catalyze the oxidation of $SO_2$ and subsequence formation of sulfuric acid or sulfate particles.

### 3.4 Whether vanadium can be used as tracers of ship fuel combustion in port area atmosphere?

### 3.4.1 Ion mass spectra difference between ships using different fuels

Owing to the fact that all samples were collected after January 1[st], 2017, the regulated date of fuel switch, it could not be judged by date whether ships had changed to diesel fuel. However, from the ion mass spectra of individual ships it could be noticed that some of the ships had higher V intensity while others not (Ni was also a certain element mostly from heavy fuel, but SPAMS seemed to perform less sensitivity of Ni detection(Agrawal et al., 2009)), and there existed considerable difference in $HSO_4^-$ intensity. In order to better distinguish ships using HFO and MDO, operation in MATLAB analysis was carried out. Signals satisfied the intersection of conditions that intensity of peak m/z=51 and m/z=67 were higher than 100 while m/z=37 lower than 100, could be recognized as vanadium-related signals, excluding the possibility of counting OC signals for V signals. Ship13 and ship17 shared a similarity in both high V and $HSO_4^-$ relative intensity, as shown in **Fig. 7**. Owing to the fact that vanadium is a typical metallic element existed mostly in heavy fuel oil with higher sulfur content than distillate fuels (Moldanov áet al., 2009;Celo et al., 2015), when obvious signal of vanadium occur in the ion mass spectra of particles from ship 13 and ship 17, they were very likely to use heavy fuel oils at berth. Accordingly, the sulfur intensity can be very high. However, for the ships with high sulfur but low vanadium, the actual contents of sulfur and vanadium of fuels used by these ships are unknown and this phenomenon cannot be well explained in this study due to the limit of instrument and the quantity of particles analyzed in each sample. The main instrument applied in this study is SPAMS, which is considered as semi-quantitative and unable to

give accurate emission factors of sulfur and vanadium. Moreover, similar studies using the same methods on shipping emissions are very rare. Therefore, this issue demands further exploration.

The two ships with vanadium signals higher than others included a total particle number of 30009 and among which 2633 were measured with ion mass spectra. In the ion mass spectra of these ships, higher $V^+$ / $VO^+$ and $HSO_4^-$ signals of over 0.8 in relative intensity while the other ships had average $HSO_4^-$ relative intensity of 0.59 and no apparent $V^+/VO^+$ signals. Due to the relatively fewer particles of such ships, there might be abnormity in low positive EC signals in their ion mass spectra. Nonetheless, major chemical PM characteristics of different fuel types could be observed through ion mass spectra.

### 3.4.2 Comparison of vanadium intensity between shipping emissions and ambient data

Previous studies have illustrated that owing to the vanadium contents in heavy fuel, vanadium in ambient atmosphere could be used as tracers indicating the heavy fuel combustion related to ship exhausts(Celo and Dabek-Zlotorzynska, 2010). Field measurements in Shanghai Port in 2011 revealed that the V/Ni ratio could be applied to identification of shipping traffic emission(Zhao et al., 2013); domestic researches elsewhere focused on Bohai Rim in 2013 identified and quantified contribution of ship PM emission to ambient air quality using V as tracers(Zhang et al., 2014). Moreover, studies in Europe also focused on metallic elements representing shipping-source particulate matters in atmosphere, which played crucial roles in ambient PM source apportionment analysis (Marco et al., 2011;Perez et al., 2016). As the new regulation came into effect and ships were demanded to use low sulfur diesel oil at berth, whether vanadium could still be used as tracers of ship emission to accomplish shipping source identification is worthy of exploration. Distinct from previous researches that mainly based on filter analysis, this study provided an innovative perspective of spectrometry analysis.

Owing to the working principle of SPAMS, mass fraction of vanadium element to other symbolic elements could not be derived, and Ni as well as La showed no apparent signals in SPAMS spectra. The main methodology was to make comparison between the vanadium intensity of particles from ships to that of particles sampled in ambient atmosphere. If the vanadium intensity in ship exhausts is commonly lower than that of the corresponding ambient data, then it is no longer a proper tracer for vanadium to identify ship exhaust sources. Calculation was performed using a method similar to the one mentioned previously. Due to the fact that sampled and particle numbers detected by SPAMS differ apparently among ships, we defined the ratio of total intensity to SPAMS detected particle number as vanadium-

related intensity. The results of overall 20 ships sampled during January 4[th] to January 16[th] as well as the ambient vanadium intensity is shown as **Fig. 8**.

Noticing that the ordinate was logarithmic, considerable distinction existed among ships in vanadium intensity. The intensities in the exhausts from 17ships were well below or comparable to corresponding ambient intensity, while ship2, ship13 and ship17 can be observed with apparently high vanadium intensity. This phenomenon might revealed the fact that after fuel substitution, vanadium from ship exhausts at berth would not appear remarkably higher than that of the ambient in most cases. Another crucial evidence is the element analysis of 3 diesel samples collected during field measurements (shown in **Table 4**). None of the 3 diesel samples were vanadium detected, which showed significantly decrease from the literature data of 38.0-133.8mg/kg in previous studies using intermediate fuel oils(IFO) as shown in **Table 4**(Celo et al., 2015). From the aspect of vanadium "sources", it can be referred that if the content of vanadium in fuel keeps declining, the role it plays will not be as significant as it used to be. It will be much harder to detect the existence of vanadium in ambient atmosphere.

### 3.5   Sulfate particle contribution of at berth shipping emissions

The primary goal of domestic ship emission control areas was to control $SO_2$ and sulfate emissions. The concrete demand for emission reduction was lowering sulfur content in fuel oil to a certain level, which had been designed as lower than 0.5% by weight percentage. Based on these facts, it's meaningful to evaluate the impact of switching to low sulfur oils on ambient air quality. One of the effective methods was to identify sulfate particles from ships out of overall ambient particles that were sampled by SPAMS and observe the change of sulfate contribution from ships.

The first step was to identify and extract ship-source particles out of all that were sampled and analyzed by SPAMS. The operation was based on information obtained from the analyzed ion mass spectra after on-board sampling. Then by algorithm calculation via MATLAB, certain particles are identified as from ship exhaust. The next was to extract sulfate particles from the identified particles from ships and ambient, respectively. This step was accomplished by finding particles as sulfates with m/z=-97 or -80, which represented $HSO_4^-$ and $SO_3^-$, two typical markers of sulfates. Finally the temporal (1h resolution) number change of identified sulfate particles from ships and ambient particles as well as the ratio was obtained. The temporal profile (1 hour resolution) of ship-source and ambient particle number sampled by SPAMS

between 18:00, December 27th and 18:00, January 15th was shown in **Fig. 9(a)**, while the green line referred to the number fraction of ship-source to ambient sulfates. The red dashed line vertical to the time axis represented the time of 0:00 on January 1st when regulation of demanding clean diesel fuel to be switched came into effect.

A total of 439 sets of data were included in the source apportionment analysis. By linear fitting for ship-source and ambient sulfate particle numbers, the Pearson Correlation Coefficient was 0.91 as shown in **Fig. 9(c)**, indicating that sulfate particles from ships and ambient had a strong correlation with each other. Meanwhile, it also can be seen from the change tendency that the change of sulfate particles in ambient air always kept synchronous with that of ship-source particles, proving the fact that ship-source sulfate particles had made relatively stable contribution to ambient sulfate particle concentration.

The analysis was divided into two sections by the time of January 1st, 2017. Before the time spot, it can be seen that around December 29th, both ambient and ship-source sulfate particles formed several peaks as well as the ship-source/ambient ratio. Peaks of ship sulfate particles were formed largely depending on ships sailing into or out of the berth, thus producing plumes sampled by SPAMS. In the following days during December 29th to January 4th, evident decrease could be seen, especially around January 1st, which might due to switching to low sulfur diesel oil. Brief surveys had been conducted towards crew on board during measurement periods and it was informed that some of the ships had switched to low sulfur oil in advance of the regulated date. Between January 4th and January 6th a peak higher than others could be observed, while the ratio kept even lower than around December 29th. The ship-source/ambient sulfate ratio reached peaks at 18:00 on January 9th and 16:00 on January 12th respectively, however, during these two time sections the quantity of measured particle numbers was apparently low, which could cause errors that were easy to be neglected. Generally, the average ratio of ship source sulfate particles to ambient sulfate particles before and after January 1st were 23.82% and 23.61%, respectively. With regard to the uncertainty in sampling, analysis and calculation, the results can be regarded as unchanged at a level of 24%.

To better focus on the shipping emissions, we take the wind direction data into consideration. The wind direction of the whole sampling period was shown as **Fig. 9(b)**. According to the geographic positions of berths and wind directions, as the berths mainly distribute in the northwest, north and east direction of the sampling site, wind from these directions will driving the plumes to the sampling site. Moreover, no obvious emission sources other than ships at berth could interfere the ambient sampling.

Ambient data with wind direction in the range of northwest to southeast (clockwise) were extracted and divided by January 1st, 2017. A total of 10 hours with 37825 particles and a total of 133 hours with 682176 particles were calculated before and after January 1st, respectively. The updated results for the ratio of sulfates identified as shipping emissions to ambient were 35% and 27% before and after January 1st, 2017 respectively, indicating a decrease of at berth shipping emission contribution to ambient sulfates. The new Chinese sulfur limit for auxiliary engines corresponds to the global shipping limit that will apply from the year 2020. Studies have been done to estimate the effect of the low sulfur fuel limit. On global level, according to Sofiev's (Sofiev et al., 2018) estimation, the global implementation of the 0.5% sulfur content policy can reduce the annual average sulfate concentration by 2-4 $\mu g/m^3$. However, even after the implementation of 0.5% sulfur limit for ships at berth, the PM and $SO_2$ emissions still remain at a level of 770 and 2500kt respectively. In China ports, under the scenario of all ships changing over to low sulfur fuel (<0.5%) in all China's emission control area, the remaining of at berth PM and $SO_2$ emissions can reach up to 1kt and 8kt respectively in Jing-Jin-Ji port area(Liu et al., 2018). If electricity from land could be applied, these emissions could be further reduced.

According to the source apportionment of sulfate particles in port area, the number concentration contribution of sulfates from shipping emission at berth are lowered from 35% to 27% after the switching oil policy implementation after January 1st, 2017. The stricter fuel sulfur limit did reduce the contribution of shipping emission, but these emissions would continue to play an important role in atmospheric pollution and electricity from land will be demanded to ameliorate this situation.

In general, PM and $SO_2$ emissions can't be eliminated by merely controlling the sulfur content of fuels, though the stricter sulfur limits is an effective way to reduce emissions. Hence the option of using electricity from land will probably help maximizing the emission reduction of $SO_2$ and PM.

## 4    Conclusions

Sampling of VOCs and particles from 20 ships at berth was conducted in winter Jingtang Port during the period of December 27th, 2016 to January 15th, 2017. Average VOC source profile of container ships was obtained, and SOA yields as well as ozone forming potential were calculated based on source profile. Comparisons were made to diesel and gasoline vehicles according to the source profile in previous researches. Secondary organic aerosol yields and ozone forming potential were 0.017±0.007g SOA/g

VOCs and $2.63\pm0.37$g $O_3$/g VOCs, respectively. Results showed that VOCs from ships tended to yield more secondary organic aerosol than diesel and gasoline vehicles, while ozone forming potential was comparable to them.

SPAMS sampling and analysis provided information of particle average ion mass spectra, manual grouping, total and specified group size distribution as well as impact of ship-source sulfate particles on ambient atmosphere. EC, ECOC and Na-rich particles were predominant sorts in overall particles, occupying over 90.7% of all particles. Size distribution indicated that most particles concentrated in the range of 0.2-1.4μm, and different sorts of particles had various distribution patterns.

The issue of vanadium as tracer element was demonstrated and conclusion was drawn that after the fuel substitution, fuel vanadium contents had been significantly lowered and vanadium from ship auxiliary engine exhausts showed no obvious excess comparing to corresponding ambient data in most cases.

After identifying and extracting ship-source sulfate particles out of ambient sulfate particles during the whole sampling period, a temporal profile with resolution of 1hour was obtained. Comparing post-January 1st data to that of December, the ratio of ship-source sulfate particles to ambient sulfate particles remain unchanged at a level of 24%. When considering the wind direction with berths at upwind, the sulfate contribution of ships at berth could be observed from 35% to 27% before and after the implementation of switching oil policy. The contribution of shipping emissions at berth to the ambient sulfates was lowered by the stricter sulfur limit in fuels.

**Acknowledgements**

This work was supported by National Natural Science Foundation of China (No. 91544110 and 41571447), Beijing Nova Program (Z181100006218077), National Key R&D Program (2016YFC0201504), Special Fund of State Key Joint Laboratory of Environment Simulation and Pollution Control (16Y02ESPCT), and National Program on Key Basic Research Project (2014CB441301).

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

**Figures**

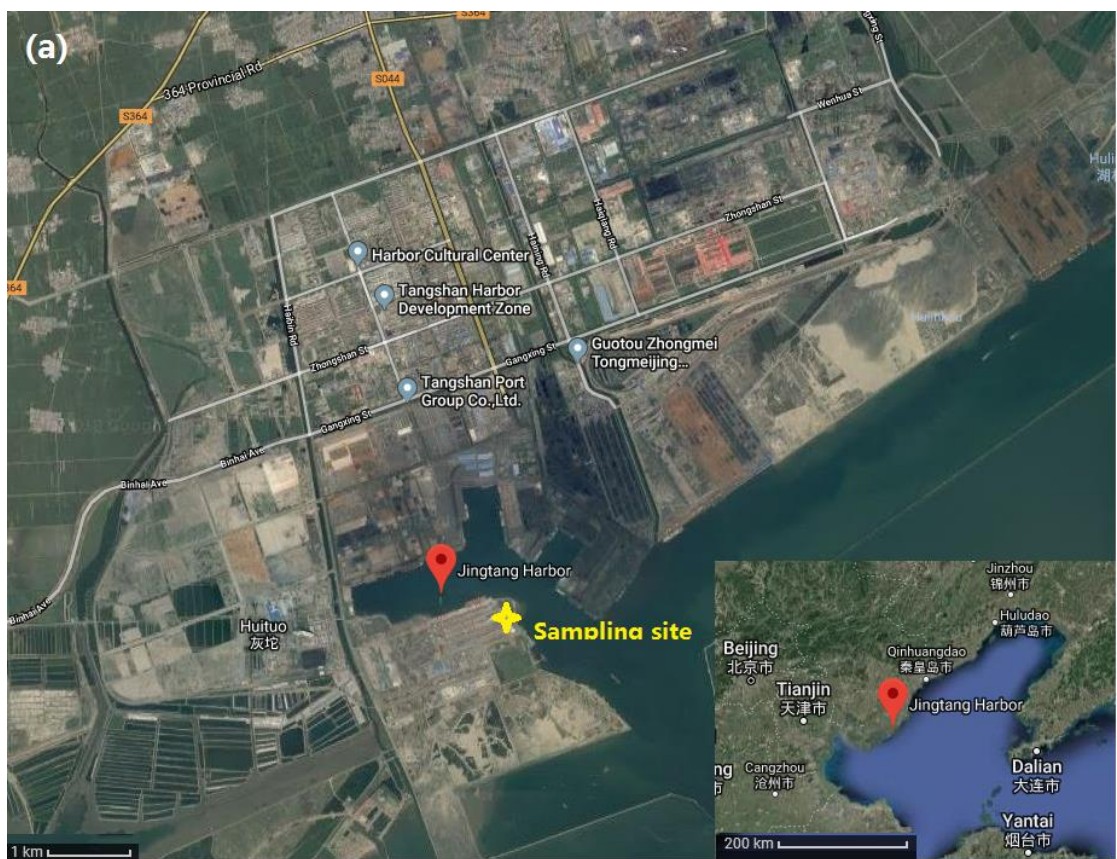

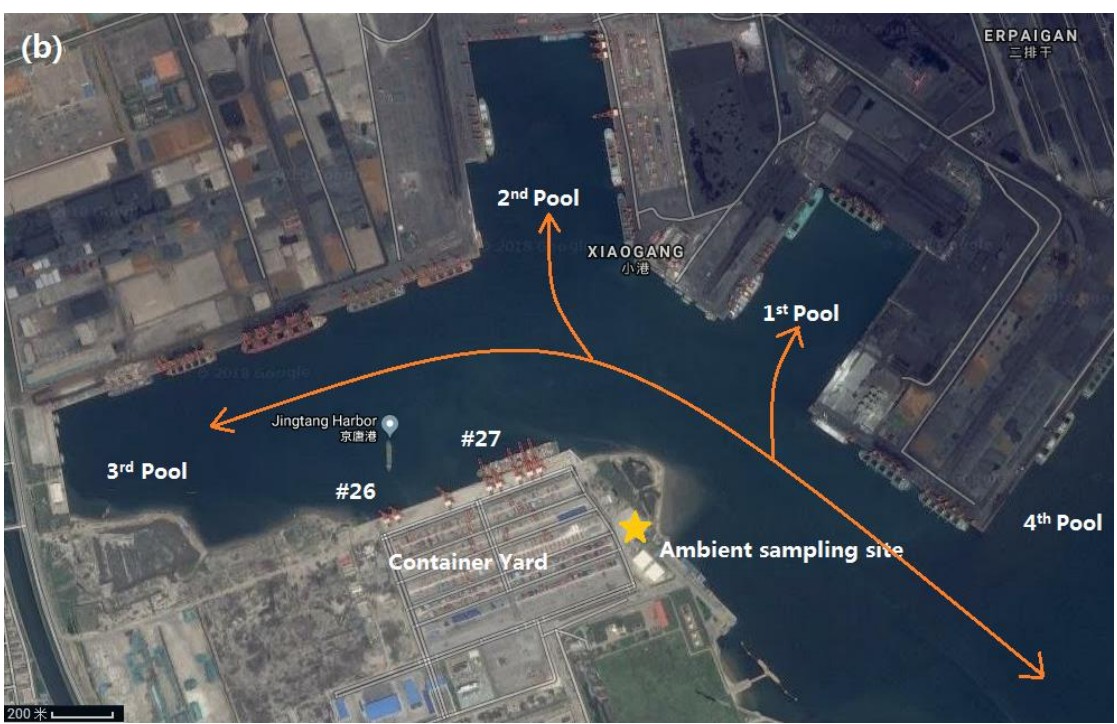

**Figure 1: (a)** The location and surroundings of Jingtang Port shown in both larger scale and smaller scale. The yellow marker represents the sampling site inside the port area. The road, residential distribution and their distance to the port can be observed by this map. **(b)** The location of sampling site on an amplified scale. The distribution of the berths, pools as well as the surroundings of the sampling site is illustrated on this map.

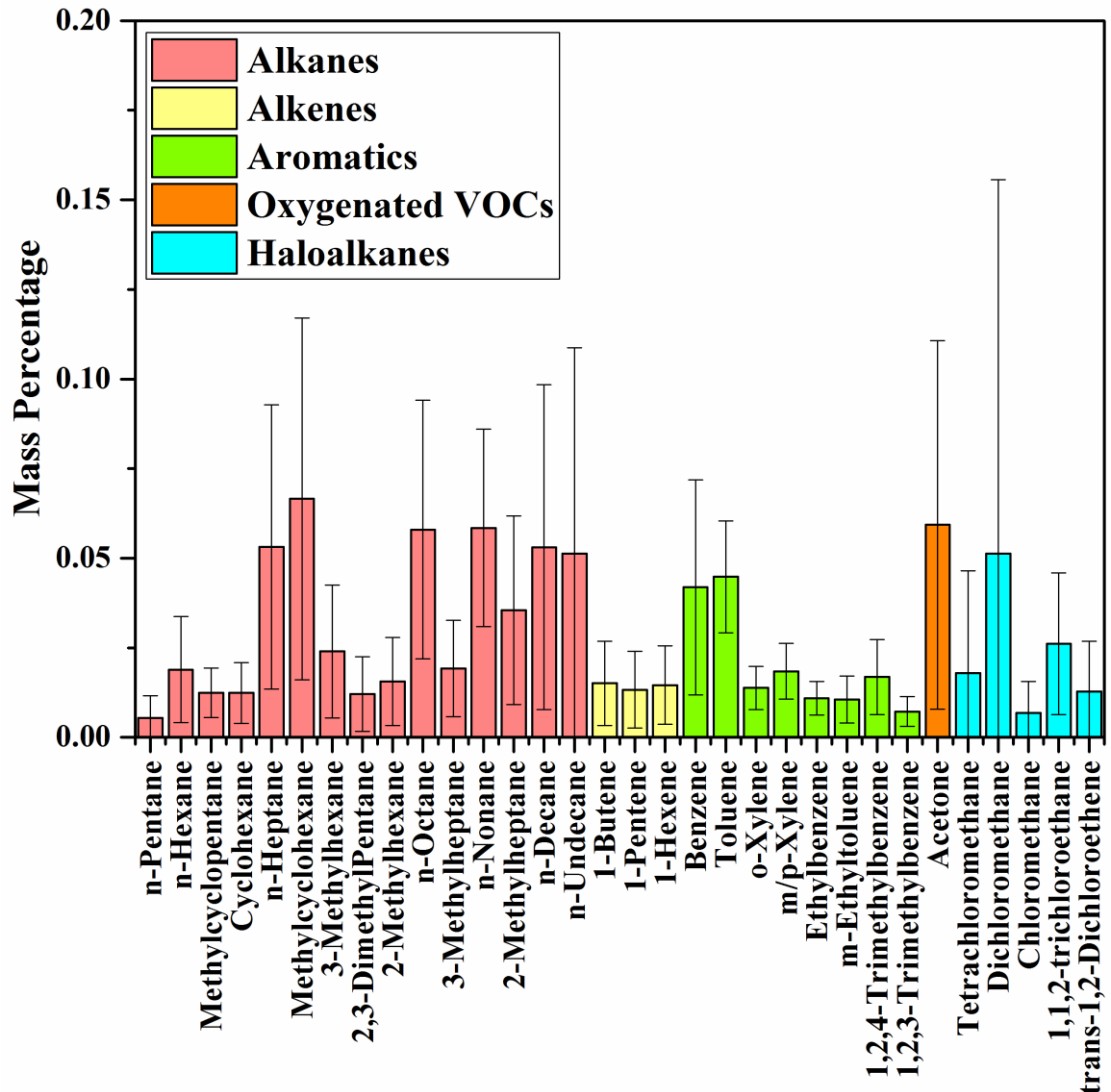

**Figure 2:** VOC source profile from ship auxiliary engine exhausts analyzed by GC-MS, 93 species including PAMS and TO-15. VOC species are classified into alkanes, alkenes, aromatics, oxygenated compounds and haloalkanes respectively. Species among top 32 by weight percentage are listed in this figure.

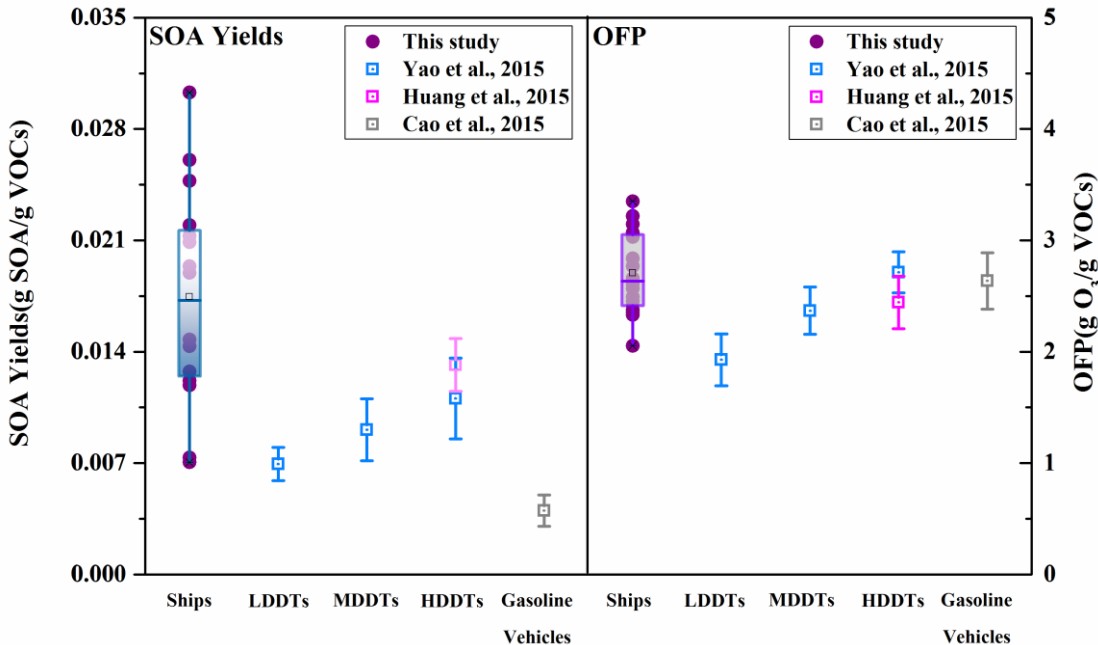

**Figure 3:** Comparison of SOA yields and ozone forming potential (OFP) calculated based on VOC source profile with calculated results from previous study (Yao et al., 2015; Cao et al., 2015; Huang et al., 2015). The purple dots represent ships tested in this study. Diesel trucks are divided into 3 types: light-duty diesel trucks (LDDTs), medium-duty diesel trucks (MDDTs) and heavy-duty diesel trucks (HDDTs).

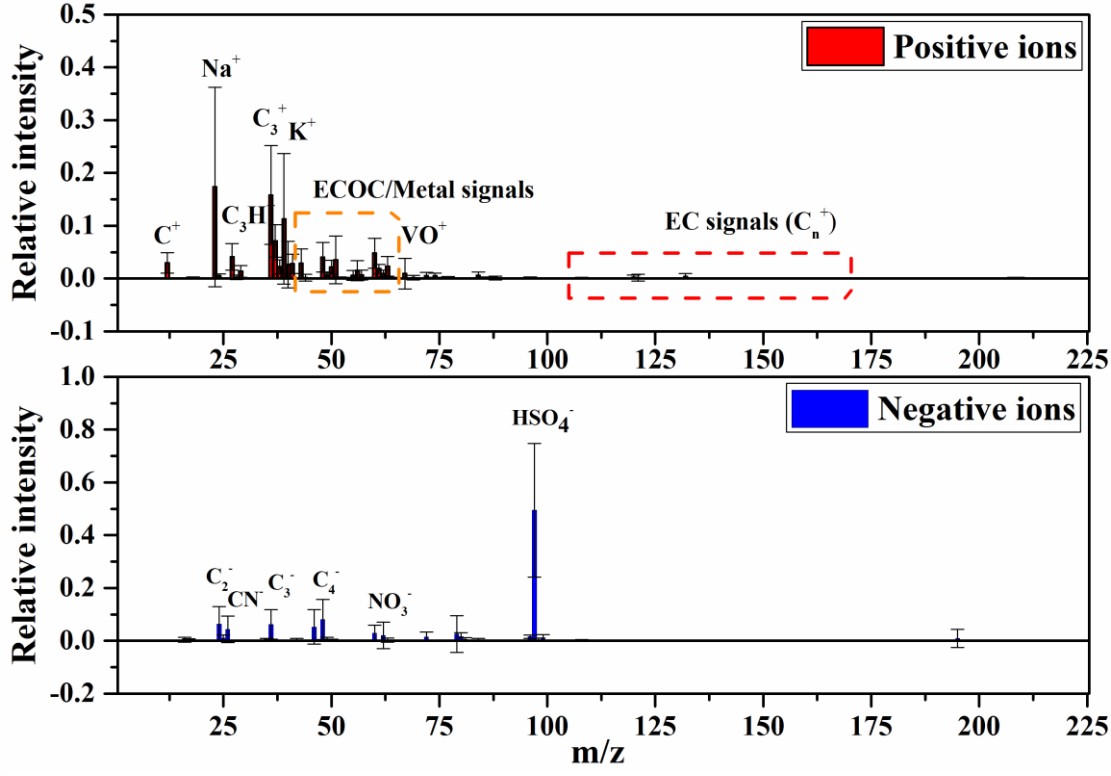

**Figure 4:** Average ion mass spectra derived from SPAMS of 20 samples of ship exhausts. Standard

deviations are given in the figure and typical ion peaks are also marked in both positive and negative ion mass spectra.

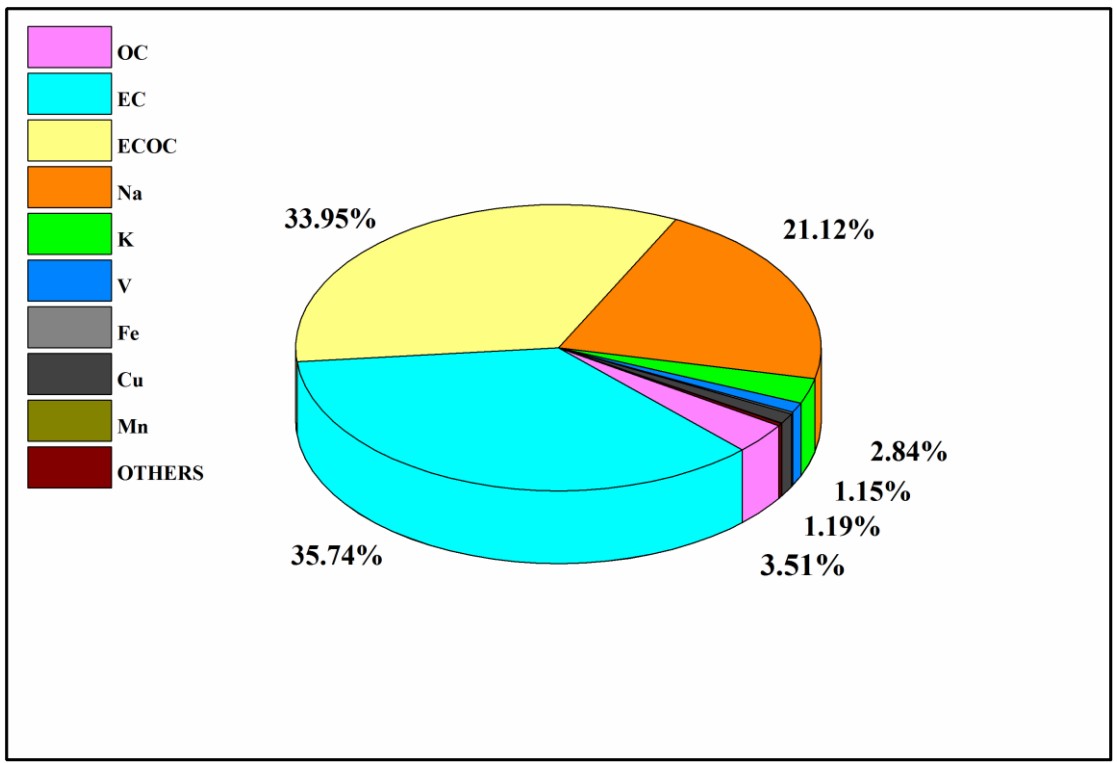

 **Figure 5:** Manual classification of ten types of particles from 20 samples and their proportion by numbers.

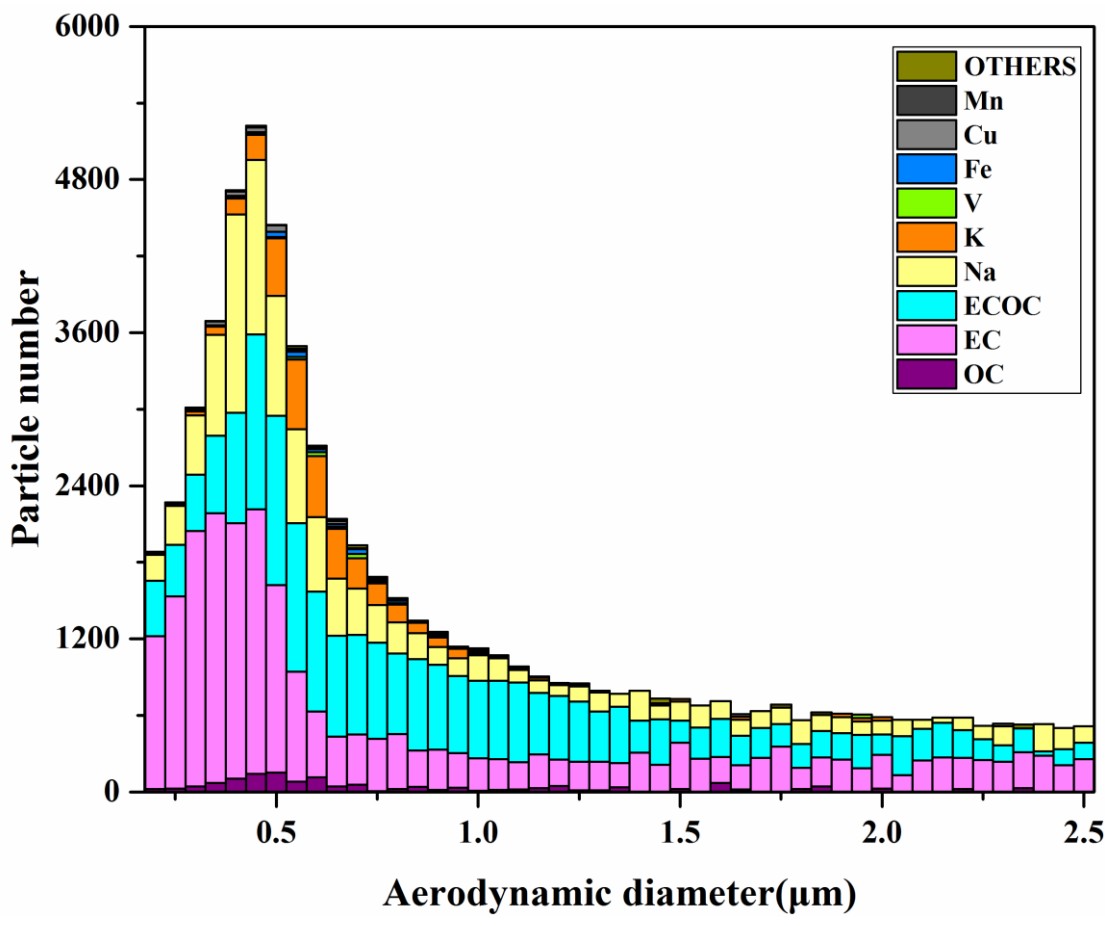

**Figure 6:** Specific size distribution of 10 classes of particles in the range of 0.2-2.5 μm.

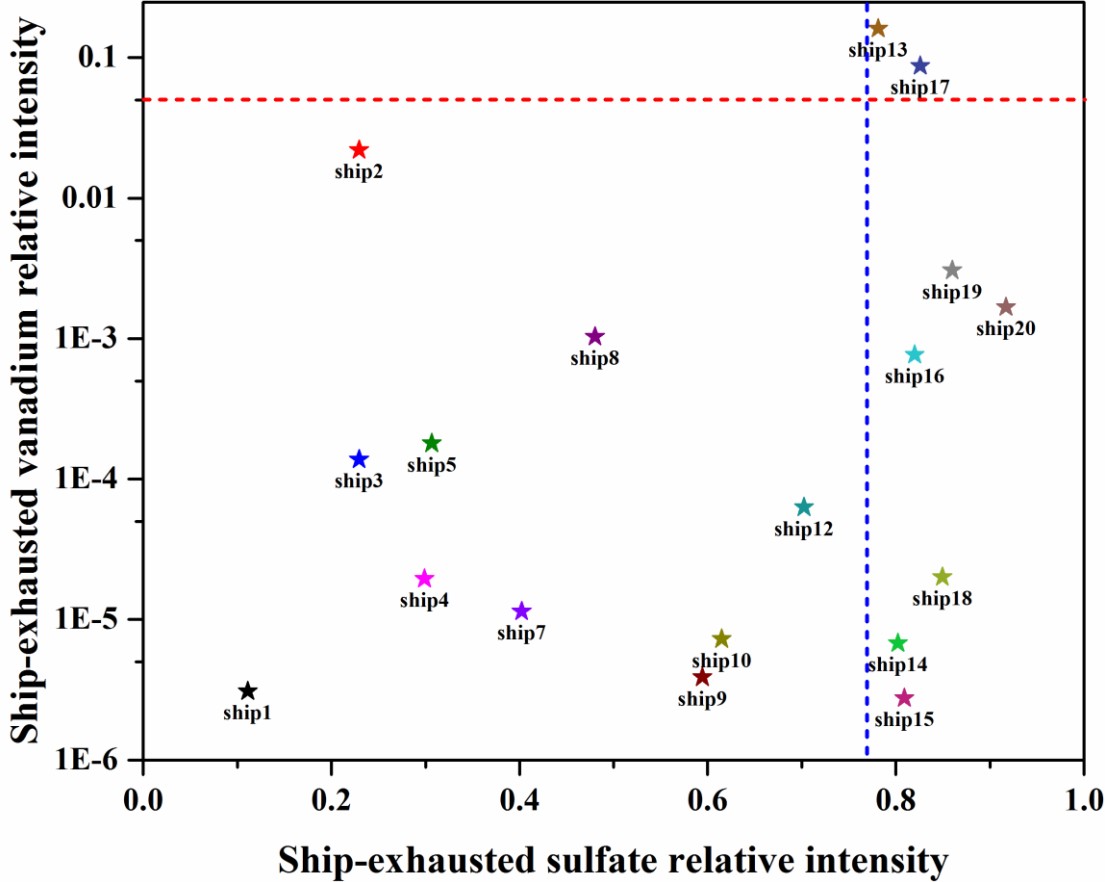

**Figure 7:** Vanadium and sulfate relative intensity correlation obtained from the ion mass spectra of 20 ships.

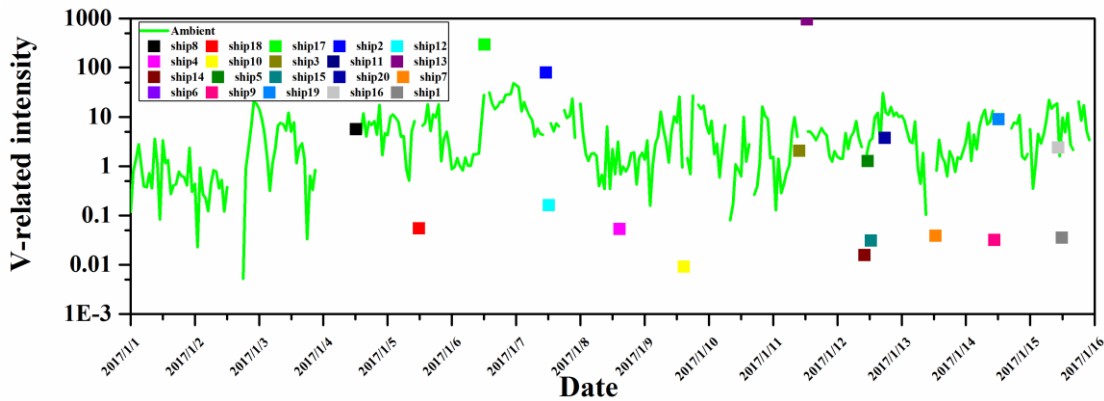

**Figure 8:** Comparison of vanadium intensity of 20 individual ship particle samples to ambient particle samples from January 1st to January 16th. The green line is the average vanadium intensity of ambient particles over time, and the colored dots represent the average vanadium intensity of different ships distributed by sampling time.

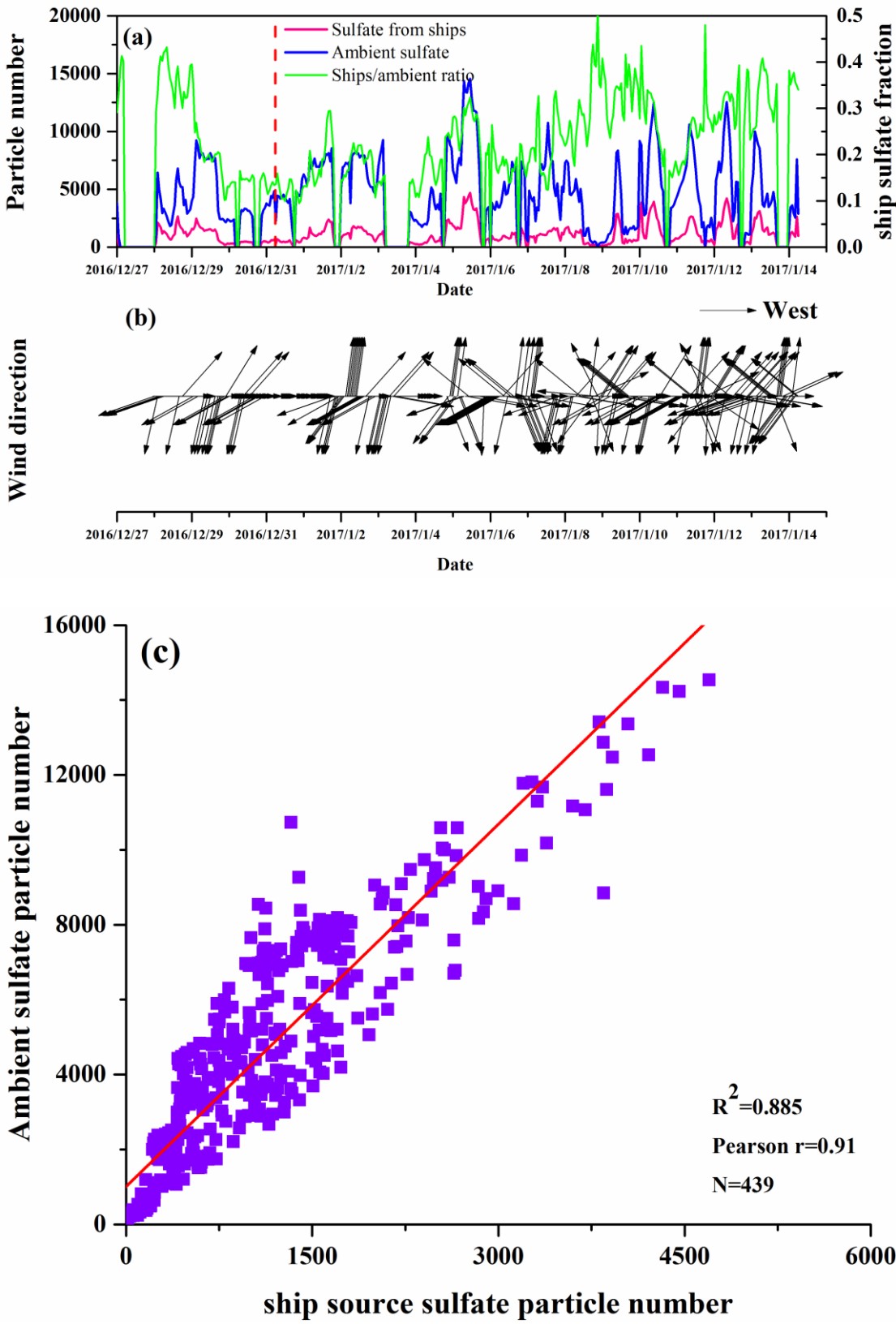

**Figure 9: (a)** Temporal number average of identified sulfate from ships, ambient sulfates and the ratio of the two numbers over time. **(b)** Wind direction variation over time during the whole sampling period. **(c)** Correlation between identified sulfates from ships and ambient sulfates. The number of data sets and

correlation coefficient are provided.

**Tables**

<table>
<tr><td colspan="2" align="center">**Table 1: Abbreviations**</td></tr>
<tr><td>**Abbreviations**</td><td>**Full name**</td></tr>
<tr><td>BC</td><td>black carbon</td></tr>
<tr><td>EC</td><td>elemental carbon</td></tr>
<tr><td>ECOC</td><td>elemental carbon-organic carbon</td></tr>
<tr><td>GC-MS</td><td>gas chromatography-mass spectrometer</td></tr>
<tr><td>HC</td><td>Hydrocarbon</td></tr>
<tr><td>HFO</td><td>heavy fuel oil</td></tr>
<tr><td>MDO</td><td>marine diesel oil</td></tr>
<tr><td>OC</td><td>organic carbon</td></tr>
<tr><td>OFP</td><td>ozone forming potential</td></tr>
<tr><td>PM</td><td>particulate matter</td></tr>
<tr><td>SOA</td><td>secondary organic aerosol</td></tr>
<tr><td>SPAMS</td><td>single particle aerosol mass spectrometer</td></tr>
<tr><td>VOCs</td><td>volatile organic compounds</td></tr>
</table>

**Table 2:** Brief information of 20 sampled ships

| Ship No. | Length*Width(m) | Model Year | Dead Weight Tonnage(t) | Auxiliary Engine Rated Power(kW) | Auxiliary Engine Rated Speed(rpm) | Sulfur Content (%) |
|---|---|---|---|---|---|---|
| 1 | 144*20.8 | 2015 | | | | |
| 2 | 255.1*37.3 | 2013 | 49717 | 1760 | 900 | |
| 3 | 140*20 | 2006 | 12301.8 | 200 | 1500 | 0.08 |
| 4 | | | | | | |
| 5 | 161*23 | | | | | |
| 6 | 147*9.8 | 2012 | | 900 | 1000 | 0.09 |
| 7 | 98*15 | 2009 | | 358.8 | 1500 | |
| 8 | | | | | | |
| 9 | 124*11.6 | 2015 | 5420 | 600 | 1500 | 0.029 |
| 10 | 158.6*22.6 | 2014 | 18060 | 900 | 1500 | 0.095 |
| 11 | 132*19 | | | | | |
| 12 | | | | 900 | | 0.02 |
| 13 | 180*28 | 2014 | 28791 | | | 0.3 |
| 14 | 158.5*22.6 | 2015 | 11872 | 900 | 1000 | 0.07 |
| 15 | 140*19.8 | 2009 | 10685 | 220 | 1000 | |
| 16 | 255.1*37.3 | 2012 | 67040 | 600 | 1500 | 0.029 |
| 17 | 180*28 | 2014 | 27821 | | | |
| 18 | | | | | | |
| 19 | 255.1*37.3 | 2013 | 66903 | 1320 | 900 | |
| 20 | | | | | | |

**Table 3:** Top 32 VOC species mass percentage from 16 container ships

| Compounds | Percentage | Standard deviation | Compounds | Percentage | Standard deviation |
|---|---|---|---|---|---|
| n-Pentane | 0.536278 | 0.629109 | 1-Pentene | 1.325768 | 1.070632 |
| n-Hexane | 1.891022 | 1.481339 | 1-Hexene | 1.457199 | 1.094036 |
| Methylcyclopentane | 1.239405 | 0.691045 | Benzene | 4.190024 | 3.000858 |
| Cyclohexane | 1.241246 | 0.847235 | Toluene | 4.47936 | 1.565238 |
| n-Heptane | 5.313712 | 3.965797 | o-Xylene | 1.379276 | 0.603842 |
| Methylcyclohexane | 6.655522 | 5.044339 | m/p-Xylene | 1.84439 | 0.778013 |
| 3-Methylhexane | 2.397949 | 1.855725 | Ethylbenzene | 1.091218 | 0.472748 |
| 2,3-DimethylPentane | 1.206319 | 1.044379 | m-Ethyltoluene | 1.05578 | 0.652482 |
| 2-Methylhexane | 1.559125 | 1.225618 | 1,2,4-Trimethylbenzene | 1.684083 | 1.045932 |
| n-Octane | 5.798276 | 3.61007 | 1,2,3-Trimethylbenzene | 0.718099 | 0.414425 |
| 3-Methylheptane | 1.918903 | 1.344047 | Acetone | 5.92993 | 5.138703 |
| n-Nonane | 5.844313 | 2.757486 | $CCl_4$ | 1.791729 | 2.854673 |
| 2-Methylheptane | 3.546112 | 2.630706 | $CH_2Cl_2$ | 5.132463 | 10.42498 |
| n-Decane | 5.305601 | 4.532773 | $CH_3Cl$ | 0.681607 | 0.871668 |
| n-Undecane | 5.125647 | 5.741735 | 1,1,2-trichloroethane | 2.610167 | 1.975113 |
| 1-Butene | 1.507865 | 1.176521 | trans-1,2-Dichloroethene | 1.274261 | 1.402993 |

**Table 4:** Element analysis of 3 fuel samples and comparison with previous studies

| | | Celo et al., 2015 | | | This study(Sampled on Jan 14th, 2017) | | |
|---|---|---|---|---|---|---|---|
| Fuel | IFO380 | IFO180 | IFO60 | MDO | MDO(ship 9) | MDO(ship 19) | MDO(unplanned ship) |
| 15℃ density(kg·$m^{-3}$) | 988 | 973.7 | 957.6 | 854.3 | 848.2 | 853.1 | 846.3 |
| w% C | 86.26 | 86.78 | 87.22 | 86.85 | 85.16 | 84.78 | 86.83 |
| w% H | 11.26 | 10.7 | 11.05 | 12.97 | 13.07 | 13.21 | 13.15 |
| w% N | 0.39 | 0.38 | 0.38 | 0.026 | 0.027 | 0.026 | 0.010 |
| w% S | 2.7 | 2.21 | 1.22 | 0.119 | 0.38 | 0.080 | 0.065 |
| mg·$kg^{-1}$ Fe | 31.44 | 17.71 | Not detected | Not detected | 2.7 | <1 | <1 |
| mg·$kg^{-1}$ V | 133.8 | 102.4 | 38.0 | Not detected | Not detected | Not detected | Not detected |
| mg·$kg^{-1}$ Ni | 63.2 | 46.5 | 21.0 | Not detected | Not detected | Not detected | Not detected |
| mg·$kg^{-1}$ Cu | 29.51 | 23.63 | Not detected | Not detected | Not detected | Not detected | Not detected |

IFO: Intermediate Fuel Oil MDO: Marine Diesel Oil

Two of the fuel samples were from ship 9 and ship 19, and the sample marked as unplanned ship indicated

that this ship was not among the 20 ships included in this study.