# Peer review of "Characteristics of marine shipping emissions at berth: profiles for PM and VOCs"

_Atmospheric Chemistry and Physics, 2017_

## Referee Comment (RC1) · Anonymous Referee #1 · 26 Feb 2018

This manuscript provides detailed information on ship atmospheric emissions while the ships are in Jingtang port and using auxiliary engines. This important topic has tended to be overlooked given the focus on emissions from ships' main engines. I recommend that authors consider the following points before publication. 1. More information is needed about Jingtang port, for example the annual traffic, exposure to other atmospheric emissions such as passing ships and centres of population. A map could be useful here. 2. It should be noted that the new Chinese sulphur limit for auxiliary engines corresponds to the global shipping limit that will apply from the year 2020. The findings presented in this manuscript are also relevant to proposals for ships in berth to be able to use electricity from land instead of auxiliary engines – it would be helpful to discuss this option in the light of the measurements made. 3. More

detail is needed concerning the ambient sampling. Was the sampling site placed so that it was significantly affected by passing ship traffic and/or built up areas and road traffic? 4. In the conclusions, a "slight decrease from 23.82% to 23.61%" is noted. Given the uncertainties involved in the sampling, it may be more reasonable to state that the ratio was unchanged at 24%, unless it can be shown that the ratio can be measured to better that 1% uncertainty. 5. Figure 2: the symbols for ship emissions should be explained. Why are there no error bars for diesel and gasoline? 6. Figure 4 is large and gives unnecessary detail. Delete this figure and show ranges and/or standard deviations in Figure 3. 7. Figure 7 shows a lack of clear correlation between sulphur and vanadium. 6 ships have high sulphur but low vanadium. 8. Figure 8 does not add important information and can be deleted 9. Figure 10 – complement with an ambient/vessel correlation. Editorial points: 1. A table of the abbreviations used should be included 2. The figure legends are too concise – they should provide a clear description of the data shown in each figure. 3. The English language needs editing

---

## Referee Comment (RC2) · Anonymous Referee #2 · 16 Mar 2018

The manuscript presents the emission factors of speciated particles and VOCs from 20 marine container ships at berth in eastern China. The secondary organic aerosol yield and ozone forming potential are estimated based on measured VOC species, and information on major chemical components and their number size distributions given. While acknowledging that such kind of study is import for regional and even global air quality assessment, I do have some concerns on the methods used in this study. I would recommend the manuscript not to be accepted unless the following issues have been well addressed.

(1) As stated in the manuscript, a total of 93 VOC species were detected and all of them have carbon numbers larger than four. Is it due to the limitation of the equipment and analyzing method used that lower carbon compounds were not detected? Or, that

is a real atmospheric phenomenon?

(2) A large amount of Na-rich particles are found in the sampled exhaust from the ships, which has been attributed to full of sea salt in the intake of ships. If so, could sea slat also have perturbations on other elements such as K, Mg, and Fe as well as OC? Would such a perturbation depend on environmental and meteorological conditions during the sampling time?

(3) As described in the manuscript, ambient particle sampling was conducted from 27 December 2016 to 15 January 2017. However, the positions of the ships and mea-surement site during the experiment are not well introduced. Since the ambient particle sampling was conducted in an open atmosphere, the local meteorological conditions should have a large influence on the experimental results. The ship plume effect needs to be investigated in detail.

(4) It is unclear and misleading to use a percentage to describe the ratio of sulfate par-ticles from shipping emissions (a flux) over ambient sulfate particles (a concentration).

---

## Author Comment (AC1) · 6 Apr 2018

The formatted version of this reply is also attached as the supplement file.

Dear Editor and Referees,

We are pleased to submit our responses to all the comments and revision for manuscript acp-2017-1132. We appreciate all the comments and suggestions that are especially helpful. All the referees' comments have been addressed carefully.

Best regards with respect,

Huan Liu, representing all authors

Response to Referee's Comments #1 1. More information is needed about Jingtang

port, for example the annual traffic, exposure to other atmospheric emissions such as passing ships and centres of population. A map could be useful here.

Response: Thanks for the helpful suggestion. Background information about Jingtang Port is necessary to help explain the significance of exploring the emission characteristics of shipping emissions at berth in this place. The basic information including atmospheric geographic location, annual throughput, traffic, and population are described in the Experimental and Methods part of the manuscript. A map is also provided as you suggested.

Revision in manuscript: (1) Page 4, Line 27: The title of Section 2.1.1 is revised as "Information of Jingtang Port, sampling site and ships". (2) Page 4, Line 28- Page 5, Line 20: "Ambient sampling site was located inside Jingtang Port, Tangshan City, Hebei Province, China. Jingtang Port is located in Bohai Bay and belongs to Port of Tangshan, which is among the core ports in domestic emission control area. According to the China Port Yearbook 2015, the annual traffic of ships in Port of Tangshan reached 15084 and the total throughput exceeded 500 million tons, ranking 5th among global port throughputs. Jingtang Port area is surrounded by the Port Economic Development Area, which has a population of 78, 300. Tangshan is a typical industrial city with average PM2.5 concentration in winter of 117 $\mu$g/m3(Zhang et al., 2017). The current PM2.5 source apportionment studies in Tangshan did not include shipping emissions due to the lack of basic information and researches. The background information indicates the significance and urgency of studying the impact of shipping emissions and the effect of the fuel switching policy. As is shown in Fig. 1(a), the center of population mainly concentrates in the residential area, located in the north of the port area, about 2km away from the port. About 2.5km in the west of the port area there is a thermal power plant with after-treatment facilities according to the strict emission control standards to power plants in China. Between the port and the other zones are two main roads with trucks driving to carry containers in and out of the port, which is about 1km away from the sampling site. Besides trucks and the power plants, there's no further

emission source near the port area. The site for ambient particle collection and instrumental analysis is surrounded by the four pools and the channel, on an open and flat corner close to the #26 and #27 Berth as well as the container yard inside the port, as is shown in Fig. 1(b). No tall buildings exist around the sampling instrument. The distribution of berths, pools and the sampling site guarantees that plumes from ships at berth are prone to reach the sampling instrument." (3) Figure 1(a) with the title of "The location and surroundings of Jingtang Port shown in both larger scale and smaller scale" is added to briefly introduce the location of Jingtang Port and its surroundings.

2. It should be noted that the new Chinese sulphur limit for auxiliary engines corresponds to the global shipping limit that will apply from the year 2020. The findings presented in this manuscript are also relevant to proposals for ships in berth to be able to use electricity from land instead of auxiliary engines – it would be helpful to discuss this option in the light of the measurements made.

Response: Yes, we agree that the results presented in this article should be combined with the discussion of the ultimate goal of lowering shipping emissions to the maximum. This issue is discussed in detail from two perspectives in the end of Section 3.5. Both the estimation of emission reduction by other studies and the results of the source apportionment in this study are used to illustrate the importance of land electricity.

Revision in Manuscript: Page 16, Line 6- Line 22: "The new Chinese sulfur limit for auxiliary engines corresponds to the global shipping limit that will apply from the year 2020. Studies have been done to estimate the effect of the low sulfur fuel limit. On global level, according to Sofiev's (Sofiev et al., 2018) estimation, the global implementation of the 0.5% sulfur content policy can reduce the annual average sulfate concentration by 2-4 $\mu$g/m3. However, even after the implementation of 0.5% sulfur limit for ships at berth, the PM and SO2 emissions still remain at a level of 770 and 2500kt respectively. In China ports, under the scenario of all ships changing over to low sulfur fuel (<0.5%) in all China's emission control area, the remaining of at berth PM and SO2 emissions can reach up to 1kt and 8kt respectively in Jing-Jin-Ji port area(Liu et al., 2018). If electricity from land could be applied, these emissions could be further reduced. According to the source apportionment of sulfate particles in port area, the number concentration contribution of sulfates from shipping emission at berth are lowered from 35% to 27% after the switching oil policy implementation after January 1st, 2017. The stricter fuel sulfur limit did reduce the contribution of shipping emission, but these emissions would continue to play an important role in atmospheric pollution and electricity from land will be demanded to ameliorate this situation. In general, PM and SO2 emissions can't be eliminated by merely controlling the sulfur content of fuels, though the stricter sulfur limits is an effective way to reduce emissions. Hence the option of using electricity from land will probably help maximizing the emission reduction of SO2 and PM."

3. More detail is needed concerning the ambient sampling. Was the sampling site placed so that it was significantly affected by passing ship traffic and/or built up areas and road traffic?

Response: Thank you for the important suggestion and information about ambient sampling site should be well introduced. It should be noticed that the collection of particles and VOCs emitted by ships at berth was accomplished via on-board sampling directly from the vessel stacks, not from the plume in the ambient sampling site. All the emission profiles were based on direct sampling from stacks. Ambient particles were sampled by SPAMS in the sampling site and the source apportionment was done using the species profiles combined with all the ambient particle information. The detailed description of the sampling site is added in Section 2.1.1 following the introduction of the port information. An amplified map is used here (Figure 1 (b)) in order to clearly illustrate the impact of ships and road traffic on the ambient sampling.

Revision in Manuscript: (1) A new Figure 1(b) is added showing the surroundings of the sampling site in the port area.

(2) Page 5, Line 16-20: "The site for ambient particle collection and instrumental analysis is surrounded by the four pools and the channel, located on an open and flat corner

close to the #26 and #27 Berth as well as the container yard inside the port, as is shown in Fig. 1(b). No tall buildings exist around the sampling instrument. The distribution of berths, pools and the sampling site guarantees that plumes from ships at berth are prone to reach the sampling instrument."

4. In the conclusions, a "slight decrease from 23.82% to 23.61%" is noted. Given the uncertainties involved in the sampling, it may be more reasonable to state that the ratio was unchanged at 24%, unless it can be shown that the ratio can be measured to better that 1% uncertainty.

Response: We consider your suggestion very reasonable. Indeed, we didn't take the uncertainty into consideration and the accuracy of sampling and instruments is not high enough to achieve uncertainty better than 1%. Therefore, if the wind direction is not considered, the sulfate number concentration contribution of shipping emissions at berth is considered as unchanged after the switching oil policy implementation during the whole sampling period. Secondly, we have done some update on the source apportionment of sulfate particles. If taking the variation of wind direction into consideration during sampling, the result is different. After combining the wind direction with the position of sampling site and berths, we selected the ambient data during periods of certain wind direction including northwest, north and conducted source apportionment using the same method. The updated result is 35% before January 1st, 2017 and 27% after January 1st, 2017, indicating a decrease of at berth shipping emission contribution to ambient sulfates.

Revision in manuscript: (1) Abstract, Page 1, Line 26-29: "The average percentage of sulfate particles from shipping emissions before and after switching to marine diesel oil kept unchanged at a level of 24%. Under certain wind direction with berths on upwind directions, the ratio before and after January 1st is 35% and 27% respectively." (2) Text, Page 15, Line 22-Page 16, Line 5: "Generally, the average ratio of ship source sulfate particles to ambient sulfate particles before and after January 1st were 23.82% and 23.61%, respectively. With regard to the uncertainty in sampling, analysis and

calculation, the results can be regarded as unchanged at a level of 24%. To better focus on the shipping emissions, we take the wind direction data into consideration. The wind direction of the whole sampling period was shown as Fig. 9(b). According to the geographic positions of berths and wind directions, as the berths mainly distribute in the northwest, north and east direction of the sampling site, wind from these directions will driving the plumes to the sampling site. Moreover, no obvious emission sources other than ships at berth could interfere the ambient sampling. Ambient data with wind direction in the range of northwest to southeast (clockwise) were extracted and divided by January 1st, 2017. A total of 10 hours with 37825 particles and a total of 133 hours with 682176 particles were calculated before and after January 1st, respectively. The results considering wind direction for the ratio of sulfates identified as shipping emissions to ambient were 35% and 27% respectively for the two periods, indicating a decrease of at berth shipping emission contribution to ambient sulfates." (3) Conclusion, Page 17, Line 13-18: "Comparing post-January 1st data to that of December, the ratio of ship-source sulfate particles to ambient sulfate particles remained unchanged at a level of 24%. When considering the wind direction with berths at upwind, the sulfate contribution of ships at berth could be observed from 35% to 27% before and after the implementation of switching oil policy. The contribution of shipping emissions at berth to the ambient sulfates was lowered by the stricter sulfur limit in fuels." (4) Previous Figure 10 has now been revised with the change of wind direction upon time as Figure 9 (a) and (b).

5. Figure 2: the symbols for ship emissions should be explained. Why are there no error bars for diesel and gasoline?

Response: Accepted. One single purple dot represented a value of SOA yield/OFP of one individual ship. The ranges and error bars are added in Figure 2 (now renamed as Figure 3). In addition, more data from related researches (Cao et al., 2015; Huang et al., 2015; Yao et al., 2015) are also collected in the revised figure to make a comprehensive comparison between this study and literature results.

Revision in manuscript: (1) Page 8, Line 15-18: "Also, VOC source profile of 3 types of diesel trucks (light-, middle- and heavy-duty truck respectively) (Yao et al., 2015;Huang et al., 2015) and profiles of heavy-duty diesel trucks in Huang's study(Huang et al., 2015) were referenced to calculate and make comparison." (2) Previous Figure 2 has been revised as Figure 3. Error bars are more data plots from Huang's study are added.

6. Figure 4 is large and gives unnecessary detail. Delete this figure and show ranges and/or standard deviations in Figure 3.

Response: Accepted. The original Figure 3 (now renamed as Figure 4) was revised by showing the ranges and standard deviations. The previous Figure 4 and corresponding contents in the manuscript have been deleted.

Revision in manuscript: (1) Page 9, Line 12: Figure 4 and the sentence" Individual ion mass spectra for each ship were shown in Fig. 4." in the text has been deleted. (2) Previous Figure 3 has been revised as Figure 4 with standard deviation and typical ion signals have been marked in this figure.

7. Figure 7 shows a lack of clear correlation between sulfur and vanadium. 6 ships have high sulfur but low vanadium.

Response: Firstly, owing to the fact that vanadium is a typical metallic element existed mostly in heavy fuel oil with higher sulfur content than distillate fuel (Moldanová et al., 2009;Celo et al., 2015), when obvious signal of vanadium occur in the ion mass spectra of particles from a certain ship, this ship is very likely to use heavy fuel oil. Accordingly, the sulfur intensity can be very high. However, for the ships with high sulfur but low vanadium, the actual contents of sulfur and vanadium of fuels used by these ships are unknown and this phenomenon cannot be well explained in this study due to the limit of instrument and the quantity of particles analyzed in each sample. The main instrument applied in this study is SPAMS, which is considered as semi-quantitative and unable to give accurate emission factors of sulfur and vanadium. Moreover, similar studies using

the same methods on shipping emissions are very rare. Therefore, this issue demands further exploration.

Revision in manuscript: Page 12, Line 21-Page 13, Line 2: "Owing to the fact that vanadium is a typical metallic element existed mostly in heavy fuel oil with higher sulfur content than distillate fuels (Moldanová et al., 2009;Celo et al., 2015), when obvious signal of vanadium occurred in the ion mass spectra of particles from ship 13 and ship 17, these ships were very likely to use heavy fuel oils at berth. Accordingly, the sulfur intensity could be very high. However, for the ships with high sulfur but low vanadium, the actual contents of sulfur and vanadium of fuels used by these ships are unknown and this phenomenon cannot be well explained in this study due to the limit of instrument and the quantity of particles analyzed in each sample. The main instrument applied in this study is SPAMS, which is considered as semi-quantitative and unable to give accurate emission factors of sulfur and vanadium. Moreover, similar studies using the same methods on shipping emissions are very rare. Therefore, this issue demands further exploration."

8. Figure 8 does not add important information and can be deleted.

Response: Your suggestion regarding Figure 8 is accepted and we has deleted this figure and revised the corresponding contents in the manuscript. Revision in manuscript: Page 13, Line 3-8: "The two ships with vanadium signals higher than others included a total particle number of 30009 and among which 2633 were measured with ion mass spectra. In the ion mass spectra of these ships, higher V+ / VO+ and HSO4- signals of over 0.8 in relative intensity while the other ships had average HSO4- relative intensity of 0.59 and no apparent V+/VO+ signals. Due to the relatively fewer particles of such ships, there might be abnormity in low positive EC signals in their ion mass spectra. Nonetheless, major chemical PM characteristics of different fuel types could be observed through ion mass spectra."

9. Figure 10 – complement with an ambient/vessel correlation.

Response: Accepted. The previous Figure 10 has been renamed of Figure 9. A scatter plot figure is attached as Figure 9 (c) providing the correlation between ambient and ship source sulfate particles. A total of 439 sets of data is presented and Pearson correlation coefficient was 0.91, indicating that those two variables have strong correlation.

Revision in manuscript: (1) Page 15, Line 5-7: "A total of 439 sets of data were included in the source apportionment analysis. By linear fitting for ship-source and ambient sulfate particle numbers, the Pearson Correlation Coefficient was 0.91 as shown in Fig. 9(c)."

10. Editorial points: (1) A table of abbreviations is provided as Table 1. Table 1: Abbreviations Abbreviations Full name BC black carbon EC elemental carbon ECOC elemental carbon-organic carbon GC-MS gas chromatography-mass spectrometer HC Hydrocarbon HFO heavy fuel oil MDO marine diesel oil OC organic carbon OFP ozone forming potential PM particulate matter SOA secondary organic aerosol SPAMS single particle aerosol mass spectrometer VOCs volatile organic compounds (2) Descriptions of each figures have been revised to provide more information and explanations. (3) The language and grammar of the whole article has been reviewed and revised carefully.

Reference Celo, V., Dabek-Zlotorzynska, E., and McCurdy, M.: Chemical Characterization of Exhaust Emissions from Selected Canadian Marine Vessels: The Case of Trace Metals and Lanthanoids, Environmental Science & Technology, 49, 5220-5226, 10.1021/acs.est.5b00127, 2015. Huang, C., Wang, H. L., Li, L., Wang, Q., Lu, Q., de Gouw, J. A., Zhou, M., Jing, S. A., Lu, J., and Chen, C. H.: VOC species and emission inventory from vehicles and their SOA formation potentials estimation in Shanghai, China, Atmospheric Chemistry and Physics, 15, 11081-11096, 10.5194/acp-15-11081-2015, 2015. Liu, H., Meng, Z.-H., Shang, Y., Lv, Z.-F., Jin, X.-X., Fu, M.-L., and He, K.-B.: Shipping emission forecasts and cost-benefit analysis of China ports and key regions' control, Environmental Pollution, 236,

49-59, https://doi.org/10.1016/j.envpol.2018.01.018, 2018. Moldanová, J., Fridell, E., Popovicheva, O., Demirdjian, B., Tishkova, V., Faccinetto, A., and Focsa, C.: Characterisation of particulate matter and gaseous emissions from a large ship diesel engine, Atmospheric Environment, 43, 2632-2641, 10.1016/j.atmosenv.2009.02.008, 2009. Sofiev, M., Winebrake, J. J., Johansson, L., Carr, E. W., Prank, M., Soares, J., Vira, J., Kouznetsov, R., Jalkanen, J. P., and Corbett, J. J.: Cleaner fuels for ships provide public health benefits with climate tradeoffs, Nature Communications, 9, 406, 2018. Yao, Z., Shen, X., Ye, Y., Cao, X., Jiang, X., Zhang, Y., and He, K.: On-road emission characteristics of VOCs from diesel trucks in Beijing, China, Atmospheric Environment, 103, 87-93, 10.1016/j.atmosenv.2014.12.028, 2015. Zhang, H., Lang, J., Wei, W., Cheng, S., and Gang, W.: Pollution Characteristics and Regional Transmission of PM_(2.5) in Tangshan, Journal of Beijing University of Technology, 2017.

Please also note the supplement to this comment:
https://www.atmos-chem-phys-discuss.net/acp-2017-1132/acp-2017-1132-AC1-supplement.pdf

[Figure]

**Fig. 1.** New Fig. 1

[Figure]

**Fig. 2.** New Fig. 3

[Figure]

**Fig. 3.** New Fig. 4

[Figure]

**Fig. 4.** New Fig. 9

---

## Author Comment (AC2) · 6 Apr 2018

A formatted version of this reply is also attached as the supplementary file.

Dear Editor and Referee,

We are pleased to submit our responses to all the comments and revision for manuscript acp-2017-1132. We appreciate all the comments and suggestions that are especially helpful. All the referees' comments have been addressed carefully.

Best regards with respect,

Huan Liu, representing all authors

Response to Referee's Comments 2 1. As stated in the manuscript, a total of 93 VOC

species were detected and all of them have carbon numbers larger than four. Is it due to the limitation of the equipment and analyzing method used that lower carbon compounds were not detected? Or, that is a real atmospheric phenomenon?

Response: Thank you for coming up with this issue. First it should be noticed that not all of the 93 detected species have carbon number smaller than 4. Only alkanes, olefins and aromatics have more than 4 carbons and some haloalkanes and oxygenated compounds detected in this study have carbon number of 1-3. This issue is mainly relevant to the analysis method and instruments. VOCs were analyzed and calibrated by GC-MS 5975/7890 according to the USEPA methods of TO-15 and PAMS. Due to the limit of chromatographic column used during the analysis period, some typical organic compounds such as aldehydes and short chain carbonyls with carbon number lower than 4 were not detected in this study. The relatively high concentration of acetone shown in the VOCs column figure is in correlation with other study(Reda et al., 2015) in which emission factors of VOCs are detected. And according to previous studies(Gentner et al., 2012), compounds with higher carbon numbers (usually higher than 6) are more relevant to secondary organic aerosol yields.

Revision in manuscript: Page 6, Line 5-8: "Owing to the limit of chromatographic column property equipped on the GC-MS, alkanes and olefins with carbon number smaller than 4 were not detected. VOCs with carbon number larger than 6 are more relevant to the yield of secondary organic aerosols (Gentner et al., 2012)."

2. A large amount of Na-rich particles are found in the sampled exhaust from the ships, which has been attributed to full of sea salt in the intake of ships. If so, could sea salt also have perturbations on other elements such as K, Mg, and Fe as well as OC? Would such a perturbation depend on environmental and meteorological conditions during the sampling time?

Response: First we admit that the opinion of attributing the peak of Na+ and the existence of Na-rich particles to the sea salt is not precise. In view of the composition of

fuel combusted by marine engine, Na does exist with a mass concentration of around 13-22mg/kg in several samples of heavy fuel oil (HFO) and lower content in marine diesel oil(Moldanová et al., 2013;Lack et al., 2009). Moldanova (Moldanová et al., 2009) analyzed particle from ship exhausts by using two-step laser mass spectrometry and Na+ was observed with obvious peak in the mass spectra. According to Sippula's (Sippula et al., 2014) research on marine diesel engine, Na can also exist in lubrication oil and be exhausted along with the lubrication and grinding process. It is widely recognized that the metal content existed in exhausted particles has strong correlation with that in fuel (Celo et al., 2015;Moldanová et al., 2013). Although the content of Na is obviously lower than that of vanadium in fuel (55-133.8mg/kg), the peaks of Na+ in ion mass spectra are mostly higher than peaks of V+/VO+ signal from exhausts. This phenomenon indicate that there are external sources of Na and one of the most possible sources is sea salt, whose typical elements are Na. In conclusion, the Na-rich particles come from a multiple source of both fuel composition and sea salt perturbation. According to literature study (O'Dowd et al., 1997;Fitzgerald, 1991), sea salt aerosol is rich mainly in inorganic elements such as Mg, K, Cl and Na. Therefore, there might exist perturbation regarding K, Mg and Na. The content of K in exhausted particles is also higher than the other elements with their contents comparable to K in fuels. The mechanism of K from is similar to that of Na. The content of Mg in marine fuels is obviously lower than other metallic elements(Moldanová et al., 2009;Sippula et al., 2014;Winnes and Fridell, 2012) (e.g. <1mg/kg in Moldanova's research) and almost no Mg+ signals are observed in the ion mass spectra because of the probable interference of C2+ signal sharing the same m/z of 24. As for Fe, it is rarely mentioned to exist widely in sea salt. Due to the fact that OC is predominant content in engine exhausts, the perturbation from sea salt OC can be ignored even if there exists such perturbation. Considering the fact that the particles were directly collected from inside the exhaust pipes, which is unlike the plume capture in the open air, the possibility of external impacts is greatly lowered. And no discussions with regard to this issue are founded in other relevant measurements.

Revision in manuscript: Page 10, Line 14-Page 15, Line 1: "In view of the composition of fuel combusted by marine engine, Na does exist with a mass concentration of around 13-22mg/kg in several samples of heavy fuel oil (HFO) and lower content in marine diesel oil(Moldanová et al., 2013;Lack et al., 2009). Moldanova (Moldanová et al., 2009) analyzed particle from ship exhausts by using two-step laser mass spectrometry and Na+ was observed with obvious peak in the mass spectra. According to Sippula's (Sippula et al., 2014) research on marine diesel engine, Na can also exist in lubrication oil and be exhausted along with the lubrication and grinding process. It is widely recognized that the metal content existed in exhausted particles has strong correlation with that in fuel (Celo et al., 2015;Moldanová et al., 2013). Although the content of Na is obviously lower than that of vanadium in fuel (55-133.8mg/kg), the peaks of Na+ in ion mass spectra are mostly higher than peaks of V+/VO+ signal in exhaust. This phenomenon indicate that there are external sources of Na and one of the most possible sources is sea salt, whose typical elements are Na. In conclusion, the Na-rich particles come from a multiple source of both fuel composition and sea salt perturbation. Only particles with relative area of Na+ obviously higher than other positive ions were counted as Na-rich particles. The same rule also adapted to K-rich particles. Owing to the fact that the content of K in most marine fuels is lower than that of Na, V and other typical metallic elements, one of the possible sources of K-rich particles is sea salt (O'Dowd et al., 1997), which is similar to the source of Na-rich particles."

3. As described in the manuscript, ambient particle sampling was conducted from 27 December 2016 to 15 January 2017. However, the positions of the ships and measurement site during the experiment are not well introduced. Since the ambient particle sampling was conducted in an open atmosphere, the local meteorological conditions should have a large influence on the experimental results. The ship plume effect needs to be investigated in detail.

Response: Your comments regarding the influence of meteorological conditions are
quite helpful and illuminating. Wind direction is recognized as the most significant meteorological conditions. Owing to the variation of wind direction and the position distribution of berths, yards and sampling sites, the contribution of each sources identified by the ambient sampling instrument may change. For example, winds from the north and north east will expose plumes from ships to the sampling site, while west wind may bring the emissions from power plant and diesel truck exhausts from the container yard next to the sampling site. Therefore, the ships plume effect needs to be discussed considering wind directions.

Revision in manuscript: (1) Page 5, Line 16-20: "The site for ambient particle collection and instrumental analysis is surrounded by the four pools and the channel, located on an open and flat corner close to the #26 and #27 Berth as well as the container yard inside the port, as is shown in Figure 1(b). No tall buildings exist around the sampling instrument. The distribution of berths, pools and the sampling site guarantees that plumes from ships at berth are prone to reach the sampling instrument." (2) Abstract, Page 1, Line 26-29: "The average percentage of sulfate particles from shipping emissions before and after switching to marine diesel oil kept unchanged at a level of 24%. Under certain wind direction with berths on upwind directions, the ratio before and after January 1st is 35% and 27% respectively." (3) Text, Page 15, Line 22-Page 16, Line 5: "Generally, the average ratio of ship source sulfate particles to ambient sulfate particles before and after January 1st were 23.82% and 23.61%, respectively. With regard to the uncertainty in sampling, analysis and calculation, the results can be regarded as unchanged at a level of 24%. To better focus on the shipping emissions, we take the wind direction data into consideration. The wind direction of the whole sampling period was shown as Fig. 9(b). According to the geographic positions of berths and wind directions, as the berths mainly distribute in the northwest, north and east direction of the sampling site, wind from these directions will driving the plumes to the sampling site. Moreover, no obvious emission sources other than ships at berth could interfere the ambient sampling. Ambient data with wind direction in the range of northwest to southeast (clockwise) were extracted and divided by January 1st, 2017. A total of 10

hours with 37825 particles and a total of 133 hours with 682176 particles were calculated before and after January 1st, respectively. The results considering wind direction for the ratio of sulfates identified as shipping emissions to ambient were 35% and 27% respectively for the two periods, indicating a decrease of at berth shipping emission contribution to ambient sulfates." (4) Conclusion, Page 17, Line 13-18: "Comparing post-January 1st data to that of December, the ratio of ship-source sulfate particles to ambient sulfate particles remained unchanged at a level of 24%. When considering the wind direction with berths at upwind, the sulfate contribution of ships at berth could be observed from 35% to 27% before and after the implementation of switching oil policy. The contribution of shipping emissions at berth to the ambient sulfates was lowered by the stricter sulfur limit in fuels." (5) Previous Figure 10 has now been revised with the change of wind direction upon time and renamed as Figure 9 (a) and (b).

4. It is unclear and misleading to use a percentage to describe the ratio of sulfate particles from shipping emissions (a flux) over ambient sulfate particles (a concentration).

Response: First we should admit that the description of methodology may be not precise enough and caused some misunderstanding. Actually the sulfate particles from shipping emissions in the Section 3.5 are not those collected via on-board sampling. The basic principle of the source apportionment in this study is to identify the ambient sulfate particles with ion mass characteristics similar to those sampled directly from the exhaust pipes. This process is achieved by neural network and iterations via MATLAB. The description of methodology is revised accordingly.

Revision in manuscript: Page 14, Line 21-27: "The first step was to identify and extract ship-source particles out of all that were sampled and analyzed by SPAMS. The identification was based on information obtained from the analyzed ion mass spectra after on-board sampling. Then by algorithm calculation via MATLAB, certain particles were identified as from ship exhaust according to the similarity to the defined characteristics. The next was to extract sulfate particles from the identified particles from ships and ambient, respectively. This step was accomplished by finding particles as sulfates with

m/z=-97 or -80, which represented HSO4- and SO3-, two typical markers of sulfates. Finally the temporal (1h resolution) number change of identified sulfate particles from ships and ambient particles as well as the ratio was obtained."

Reference Celo, V., Dabek-Zlotorzynska, E., and McCurdy, M.: Chemical Characterization of Exhaust Emissions from Selected Canadian Marine Vessels: The Case of Trace Metals and Lanthanoids, Environmental Science & Technology, 49, 5220-5226, 10.1021/acs.est.5b00127, 2015. Fitzgerald, J. W.: Marine aerosols: A review, Atmospheric Environment.part A.general Topics, 25, 533-545, 1991. Gentner, D. R., Isaacman, G., Worton, D. R., Chan, A. W. H., Dallmann, T. R., Davis, L., Liu, S., Day, D. A., Russell, L. M., Wilson, K. R., Weber, R., Guha, A., Harley, R. A., and Goldstein, A. H.: Elucidating secondary organic aerosol from diesel and gasoline vehicles through detailed characterization of organic carbon emissions, Proceedings of the National Academy of Sciences of the United States of America, 109, 18318-18323, 10.1073/pnas.1212272109, 2012. Lack, D. A., Corbett, J. J., Onasch, T., Lerner, B., Massoli, P., Quinn, P. K., Bates, T. S., Covert, D. S., Coffman, D., and Sierau, B.: Particulate emissions from commercial shipping: Chemical, physical, and optical properties, Journal of Geophysical Research Atmospheres, 114, doi:10.1029/2008JD011300, 2009. Moldanová, J., Fridell, E., Popovicheva, O., Demirdjian, B., Tishkova, V., Faccinetto, A., and Focsa, C.: Characterisation of particulate matter and gaseous emissions from a large ship diesel engine, Atmospheric Environment, 43, 2632-2641, 2009. Moldanová, J., Fridell, E., Winnes, H., and Holminfridell, S.: Physical and chemical characterisation of PM emissions from two ships operating in European Emission Control Areas, Atmospheric Measurement Techniques, 6, 3577-3596, 2013. O'Dowd, C. D., Smith, M. H., Consterdine, I. E., and Lowe, J. A.: Marine aerosol, sea-salt, and the marine sulphur cycle: a short review, Atmospheric Environment, 31, 73-80, 1997. Reda, A. A., Schnelle-Kreis, J., Orasche, J., Abbaszade, G., Lintelmann, J., Arteaga-Salas, J. M., Stengel, B., Rabe, R., Harndorf, H., and Sippula, O.: Gas phase carbonyl compounds in ship emissions: Differences between diesel fuel and heavy fuel oil operation, Atmospheric Environment, 112, 370-380, 2015. Sippula, O.,

Stengel, B., Sklorz, M., Streibel, T., Rabe, R., Orasche, J., Lintelmann, J., Michalke, B., Abbaszade, G., and Radischat, C.: Particle emissions from a marine engine: chemical composition and aromatic emission profiles under various operating conditions, Environmental Science & Technology, 48, 11721-11729, 2014. Winnes, H., and Fridell, E.: Particle Emissions from Ships: Dependence on Fuel Type, Journal of the Air & Waste Management Association, 59, 1391-1398, 10.3155/1047-3289.59.12.1391, 2012.

Please also note the supplement to this comment:
https://www.atmos-chem-phys-discuss.net/acp-2017-1132/acp-2017-1132-AC2-supplement.pdf
* * *
[Figure]

Fig. 1. New Fig. 1

**(a)**
Particle number / ship sulfate fraction chart with legend:
— Sulfate from ships
— Ambient sulfate
— Ships/ambient ratio

**(b)** Wind direction — West

**(c)** Ambient sulfate particle number vs Ship source sulfate particle number

Pearson r=0.909
N=439

**Fig. 2.** New Fig. 9

---

## Author Comment (AC3) · 6 Apr 2018

See supplement.

Please also note the supplement to this comment:
https://www.atmos-chem-phys-discuss.net/acp-2017-1132/acp-2017-1132-AC3-supplement.pdf

---

## Author Response (AR1)

**Response to Referee's Comments #1**

**1. More information is needed about Jingtang port, for example the annual traffic, exposure to other atmospheric emissions such as passing ships and centres of population. A map could be useful here.**

**Response:**

Thanks for the helpful suggestion. Background information about Jingtang Port is necessary to help explain the significance of exploring the emission characteristics of shipping emissions at berth in this place. The basic information including atmospheric geographic location, annual throughput, traffic, and population are described in the Experimental and Methods part of the manuscript. A map is also provided as you suggested.

**Revision in manuscript:**

*(1) Page 4, Line 27: The title of Section 2.1.1 is revised as "Information of Jingtang Port, sampling site and ships".*

*(2) Page 4, Line 28- Page 5, Line 20: "Ambient sampling site was located inside Jingtang Port, Tangshan City, Hebei Province, China. Jingtang Port is located in Bohai Bay and belongs to Port of Tangshan, which is among the core ports in domestic emission control area. According to the China Port Yearbook 2015, the annual traffic of ships in Port of Tangshan reached 15084 and the total throughput exceeded 500 million tons, ranking 5$^{th}$ among global port throughputs. Jingtang Port area is surrounded by the Port Economic Development Area, which has a population of 78, 300. Tangshan is a typical industrial city with average PM$_{2.5}$ concentration in winter of 117 μg/m$^3$(Zhang et al., 2017). The current PM$_{2.5}$ source apportionment studies in Tangshan did not include shipping emissions due to the lack of basic information and researches. The background information indicates the significance and urgency of studying the impact of shipping emissions and the effect of the fuel switching policy.*

*As is shown in **Fig. 1(a)**, the center of population mainly concentrates in the residential area, located in the north of the port area, about 2km away from the port. About 2.5km in the west of the port area there is a thermal power plant with after-treatment facilities according to the strict emission control standards to power plants in China. Between the port and the other zones are two main roads with trucks driving to carry containers in and out of the port, which is about 1km away from the sampling site. Besides trucks and the power plants, there's no further emission source near the port area.*

*The site for ambient particle collection and instrumental analysis is surrounded by the four pools and the channel, on an open and flat corner close to the #26 and #27 Berth as well as the container yard inside the port, as is shown in **Fig. 1(b)**. No tall buildings exist around the sampling instrument. The distribution of berths, pools and the sampling site guarantees that plumes from ships at berth are prone to reach the sampling instrument."*

*(3) **Figure 1(a)** with the title of "The location and surroundings of Jingtang Port shown in both larger scale and smaller scale" is added to briefly introduce the location of Jingtang Port and its surroundings.*

[Figure]

**2.** **It should be noted that the new Chinese sulphur limit for auxiliary engines corresponds to the global shipping limit that will apply from the year 2020. The findings presented in this manuscript are also relevant to proposals for ships in berth to be able to use electricity from land instead of auxiliary engines – it would be helpful to discuss this option in the light of the measurements made.**

**Response:**

Yes, we agree that the results presented in this article should be combined with the discussion of the ultimate goal of lowering shipping emissions to the maximum. This issue is discussed in detail from two perspectives in the end of Section 3.5. Both the estimation of emission reduction by other studies and the results of the source apportionment in this study are used to illustrate the importance of land electricity.

**Revision in Manuscript:**

*Page 16, Line 6- Line 22: "The new Chinese sulfur limit for auxiliary engines corresponds to the global shipping limit that will apply from the year 2020. Studies have been done to estimate the effect of the low sulfur fuel limit. On global level, according to Sofiev's (Sofiev et al., 2018) estimation, the global implementation of the 0.5% sulfur content policy can reduce the annual average sulfate concentration by 2-4 μg/m³. However, even after the implementation of 0.5% sulfur limit for ships at berth, the PM and $SO_2$ emissions still remain at a level of 770 and 2500kt respectively. In China ports, under the scenario of all ships changing over to low sulfur fuel (<0.5%) in all China's emission control area, the remaining of at berth PM and $SO_2$ emissions can reach up to 1kt and 8kt respectively in Jing-Jin-Ji port area(Liu et al., 2018). If electricity from land could be applied, these emissions could be further reduced.*

*According to the source apportionment of sulfate particles in port area, the number concentration contribution of sulfates from shipping emission at berth are lowered from 35% to 27% after the switching oil policy implementation after January 1st, 2017. The stricter fuel sulfur limit did reduce the contribution of shipping emission, but these emissions would continue to play an important role in atmospheric pollution and electricity from land will be demanded to ameliorate this situation. In general, PM and SO$_2$ emissions can't be eliminated by merely controlling the sulfur content of fuels, though the stricter sulfur limits is an effective way to reduce emissions. Hence the option of using electricity from land will probably help maximizing the emission reduction of SO$_2$ and PM."*

3. **More detail is needed concerning the ambient sampling. Was the sampling site placed so that it was significantly affected by passing ship traffic and/or built up areas and road traffic?**

**Response**:

Thank you for the important suggestion and information about ambient sampling site should be well introduced. It should be noticed that the collection of particles and VOCs emitted by ships at berth was accomplished via on-board sampling directly from the vessel stacks, not from the plume in the ambient sampling site. All the emission profiles were based on direct sampling from stacks. Ambient particles were sampled by SPAMS in the sampling site and the source apportionment was done using the species profiles combined with all the ambient particle information. The detailed description of the sampling site is added in Section 2.1.1 following the introduction of the port information. An amplified map is used here (**Figure 1 (b)**) in order to clearly illustrate the impact of ships and road traffic on the ambient sampling.

**Revision in Manuscript:**

*(1) A new **Figure 1(b)** is added showing the surroundings of the sampling site in the port area.*

[Figure]

*(2) Page 5, Line 16-20: "The site for ambient particle collection and instrumental analysis is surrounded by the four pools and the channel, located on an open and flat corner close to the #26 and #27 Berth as well as the container yard inside the port, as is shown in **Fig. 1(b)**. No*

*tall buildings exist around the sampling instrument. The distribution of berths, pools and the sampling site guarantees that plumes from ships at berth are prone to reach the sampling instrument."*

4. **In the conclusions, a "slight decrease from 23.82% to 23.61%" is noted. Given the uncertainties involved in the sampling, it may be more reasonable to state that the ratio was unchanged at 24%, unless it can be shown that the ratio can be measured to better that 1% uncertainty.**

**Response**:

We consider your suggestion very reasonable. Indeed, we didn't take the uncertainty into consideration and the accuracy of sampling and instruments is not high enough to achieve uncertainty better than 1%. Therefore, if the wind direction is not considered, the sulfate number concentration contribution of shipping emissions at berth is considered as unchanged after the switching oil policy implementation during the whole sampling period.

Secondly, we have done some update on the source apportionment of sulfate particles. If taking the variation of wind direction into consideration during sampling, the result is different. After combining the wind direction with the position of sampling site and berths, we selected the ambient data during periods of certain wind direction including northwest, north and conducted source apportionment using the same method. The updated result is 35% before January 1st, 2017 and 27% after January 1st, 2017, indicating a decrease of at berth shipping emission contribution to ambient sulfates.

**Revision in manuscript:**

*(1) Abstract, Page 1, Line 26-29: "The average percentage of sulfate particles from shipping emissions before and after switching to marine diesel oil kept unchanged at a level of 24%. Under certain wind direction with berths on upwind directions, the ratio before and after January 1st is 35% and 27% respectively."*

*(2) Text, Page 15, Line 22-Page 16, Line 5: "Generally, the average ratio of ship source sulfate particles to ambient sulfate particles before and after January 1st were 23.82% and 23.61%, respectively. With regard to the uncertainty in sampling, analysis and calculation, the results can be regarded as unchanged at a level of 24%.*

*To better focus on the shipping emissions, we take the wind direction data into consideration. The wind direction of the whole sampling period was shown as **Fig. 9(b)**. According to the geographic positions of berths and wind directions, as the berths mainly distribute in the northwest, north and east direction of the sampling site, wind from these directions will driving the plumes to the sampling site. Moreover, no obvious emission sources other than ships at berth could interfere the ambient sampling.*

*Ambient data with wind direction in the range of northwest to southeast (clockwise) were extracted and divided by January 1st, 2017. A total of 10 hours with 37825 particles and a total of 133 hours with 682176 particles were calculated before and after January 1st, respectively. The results considering wind direction for the ratio of sulfates identified as shipping emissions to ambient were 35% and 27% respectively for the two periods, indicating a decrease of at berth shipping emission contribution to ambient sulfates."*

*(3) Conclusion, Page 17, Line 13-18: "Comparing post-January 1st data to that of December, the ratio of ship-source sulfate particles to ambient sulfate particles remained unchanged at a level of 24%. When considering the wind direction with berths at upwind, the sulfate contribution of*

*ships at berth could be observed from 35% to 27% before and after the implementation of switching oil policy. The contribution of shipping emissions at berth to the ambient sulfates was lowered by the stricter sulfur limit in fuels."*

*(4) Previous **Figure 10** has now been revised with the change of wind direction upon time as **Figure 9 (a) and (b)**.*

[Figure]

5. **Figure 2: the symbols for ship emissions should be explained. Why are there no error bars for diesel and gasoline?**

**Response:**

Accepted. One single purple dot represented a value of SOA yield/OFP of one individual ship. The ranges and error bars are added in **Figure 2** (now renamed as **Figure 3**). In addition, more data from related researches (Cao et al., 2015; Huang et al., 2015; Yao et al., 2015) are also collected in the revised figure to make a comprehensive comparison between this study and literature results.

**Revision in manuscript:**

*(1) Page 8, Line 15-18: "Also, VOC source profile of 3 types of diesel trucks (light-, middle- and heavy-duty truck respectively) (Yao et al., 2015;Huang et al., 2015a) and profiles of heavy-duty diesel trucks in Huang's study(Huang et al., 2015a) were referenced to calculate and make comparison."*

*(2) Previous **Figure 2** has been revised as **Figure 3**. Error bars are more data plots from Huang's study are added.*

[Figure]

**6. Figure 4 is large and gives unnecessary detail. Delete this figure and show ranges and/or standard deviations in Figure 3.**

**Response:** Accepted. The original **Figure 3** (now renamed as **Figure 4**) was revised by showing the ranges and standard deviations. The previous **Figure 4** and corresponding contents in the manuscript have been deleted.

**Revision in manuscript:**

*(1) Page 9, Line 12: **Figure 4** and the sentence" Individual ion mass spectra for each ship were shown in Fig. 4." in the text has been deleted.*

*(2) Previous **Figure 3** has been revised as **Figure 4** with standard deviation and typical ion signals have been marked in this figure.*

[Figure]

**7. Figure 7 shows a lack of clear correlation between sulfur and vanadium. 6 ships have high sulfur but low vanadium.**

**Response:**

Firstly, owing to the fact that vanadium is a typical metallic element existed mostly in heavy fuel oil with higher sulfur content than distillate fuel (Moldanová et al., 2009;Celo et al., 2015), when obvious signal of vanadium occur in the ion mass spectra of particles from a certain ship, this ship is very likely to use heavy fuel oil. Accordingly, the sulfur intensity can be very high. However, for the ships with high sulfur but low vanadium, the actual contents of sulfur and vanadium of fuels used by these ships are unknown and this phenomenon cannot be well explained in this study due to the limit of instrument and the quantity of particles analyzed in each sample. The main instrument applied in this study is SPAMS, which is considered as semi-quantitative and unable to give accurate emission factors of sulfur and vanadium. Moreover, similar studies using the same methods on shipping emissions are very rare. Therefore, this issue demands further exploration.

**Revised in manuscript:**

*Page 12, Line 21-Page 13, Line 2: "Owing to the fact that vanadium is a typical metallic element existed mostly in heavy fuel oil with higher sulfur content than distillate fuels (Moldanová et al., 2009;Celo et al., 2015), when obvious signal of vanadium occurred in the ion mass spectra of particles from ship 13 and ship 17, these ships were very likely to use heavy fuel oils at berth. Accordingly, the sulfur intensity could be very high. However, for the ships with high sulfur but low vanadium, the actual contents of sulfur and vanadium of fuels used by these ships are unknown and this phenomenon cannot be well explained in this study due to the limit of instrument and the quantity of particles analyzed in each sample. The main instrument applied in this study is SPAMS, which is considered as semi-quantitative and unable to give accurate emission factors of sulfur and vanadium. Moreover, similar studies using the same methods on shipping emissions are very rare. Therefore, this issue demands further exploration."*

**8. Figure 8 does not add important information and can be deleted.**

**Response:**

Your suggestion regarding **Figure 8** is accepted and we has deleted this figure and revised the corresponding contents in the manuscript.

**Revision in manuscript:**

*Page 13, Line 3-8: "The two ships with vanadium signals higher than others included a total particle number of 30009 and among which 2633 were measured with ion mass spectra. In the ion mass spectra of these ships, higher $V^+ / VO^+$ and $HSO_4^-$ signals of over 0.8 in relative intensity while the other ships had average $HSO_4^-$ relative intensity of 0.59 and no apparent $V^+/VO^+$ signals. Due to the relatively fewer particles of such ships, there might be abnormity in low positive EC signals in their ion mass spectra. Nonetheless, major chemical PM characteristics of different fuel types could be observed through ion mass spectra."*

**9. Figure 10 – complement with an ambient/vessel correlation.**

**Response:**

Accepted. The previous **Figure 10** has been renamed of **Figure 9**. A scatter plot figure is attached as **Figure 9 (c)** providing the correlation between ambient and ship source sulfate particles. A total of 439 sets of data is presented and Pearson correlation coefficient was 0.91, indicating that those two variables have strong correlation.

**Revision in manuscript:**

*(1) Page 15, Line 5-7: "A total of 439 sets of data were included in the source apportionment analysis. By linear fitting for ship-source and ambient sulfate particle numbers, the Pearson Correlation Coefficient was 0.91 as shown in **Fig. 9(c)**."*

[Figure]

**10. Editorial points:**

**(1)** A table of abbreviations is provided as Table 1.

**Table 1: Abbreviations**

| Abbreviations | Full name |
| --- | --- |
| BC | black carbon |
| EC | elemental carbon |
| ECOC | elemental carbon-organic carbon |
| GC-MS | gas chromatography-mass spectrometer |
| HC | Hydrocarbon |
| HFO | heavy fuel oil |
| MDO | marine diesel oil |
| OC | organic carbon |
| OFP | ozone forming potential |
| PM | particulate matter |
| SOA | secondary organic aerosol |
| SPAMS | single particle aerosol mass spectrometer |
| VOCs | volatile organic compounds |

**(2)** Descriptions of each figures have been revised to provide more information and explanations.

**(3)** The language and grammar of the whole article has been reviewed and revised carefully.

*(3) Text, Page 15, Line 22-Page 16, Line 5: "Generally, the average ratio of ship source sulfate particles to ambient sulfate particles before and after January 1st were 23.82% and 23.61%, respectively. With regard to the uncertainty in sampling, analysis and calculation, the results can be regarded as unchanged at a level of 24%.*

*To better focus on the shipping emissions, we take the wind direction data into consideration. The wind direction of the whole sampling period was shown as **Fig. 9(b)**. According to the geographic positions of berths and wind directions, as the berths mainly distribute in the northwest, north and east direction of the sampling site, wind from these directions will driving the plumes to the sampling site. Moreover, no obvious emission sources other than ships at berth could interfere the ambient sampling.*

*Ambient data with wind direction in the range of northwest to southeast (clockwise) were extracted and divided by January 1st, 2017. A total of 10 hours with 37825 particles and a total of 133 hours with 682176 particles were calculated before and after January 1st, respectively. The results considering wind direction for the ratio of sulfates identified as shipping emissions to ambient were 35% and 27% respectively for the two periods, indicating a decrease of at berth shipping emission contribution to ambient sulfates."*

*(4) Conclusion, Page 17, Line 13-18: "Comparing post-January 1st data to that of December, the ratio of ship-source sulfate particles to ambient sulfate particles remained unchanged at a level of 24%. When considering the wind direction with berths at upwind, the sulfate contribution of ships at berth could be observed from 35% to 27% before and after the implementation of switching oil policy. The contribution of shipping emissions at berth to the ambient sulfates was lowered by the stricter sulfur limit in fuels."*

*(5) Previous **Figure 10** has now been revised with the change of wind direction upon time and renamed as **Figure 9 (a) and (b)**.*

[Figure]

**4. It is unclear and misleading to use a percentage to describe the ratio of sulfate particles from shipping emissions (a flux) over ambient sulfate particles (a concentration).**

**Response:** First we should admit that the description of methodology may be not precise enough and caused some misunderstanding. Actually the sulfate particles from shipping emissions in the Section 3.5 are not those collected via on-board sampling. The basic principle of the source apportionment in this study is to identify the ambient sulfate particles with ion mass characteristics similar to those sampled directly from the exhaust pipes. This process is achieved by neural network and iterations via MATLAB. The description of methodology is revised accordingly.

**Revision in manuscript:**

*Page 14, Line 21-27: "The first step was to identify and extract ship-source particles out of all that were sampled and analyzed by SPAMS. The identification was based on information obtained from the analyzed ion mass spectra after on-board sampling. Then by algorithm calculation via MATLAB, certain particles were identified as from ship exhaust according to the similarity to the defined characteristics. The next was to extract sulfate particles from the identified particles from ships and ambient, respectively. This step was accomplished by finding particles as sulfates with m/z=-97 or -80, which represented $HSO_4^-$ and $SO_3^-$, two typical markers of sulfates. Finally the temporal (1h resolution) number change of identified sulfate particles from ships and ambient particles as well as the ratio was obtained."*

[revised manuscript text omitted]

**Tables**

**Table 1: Abbreviations**

| Abbreviations | Full name |
|---|---|
| BC | black carbon |
| EC | elemental carbon |
| ECOC | elemental carbon-organic carbon |
| GC-MS | gas chromatography-mass spectrometer |
| HC | Hydrocarbon |
| HFO | heavy fuel oil |
| MDO | marine diesel oil |
| OC | organic carbon |
| OFP | ozone forming potential |
| PM | particulate matter |
| SOA | secondary organic aerosol |
| SPAMS | single particle aerosol mass spectrometer |
| VOCs | volatile organic compounds |

| Ship No. | Length*Width(m) | Model Year | Dead Weight Tonnage(t) | Auxiliary Engine | | Sulfur Content (%) |
|---|---|---|---|---|---|---|
| | | | | Rated Power(kW) | Rated Speed(rpm) | |
| 1 | 144*20.8 | 2015 | | | | |
| 2 | 255.1*37.3 | 2013 | 49717 | 1760 | 900 | |
| 3 | 140*20 | 2006 | 12301.8 | 200 | 1500 | 0.08 |
| 4 | | | | | | |
| 5 | 161*23 | | | | | |
| 6 | 147*9.8 | 2012 | | 900 | 1000 | 0.09 |
| 7 | 98*15 | 2009 | | 358.8 | 1500 | |
| 8 | | | | | | |
| 9 | 124*11.6 | 2015 | 5420 | 600 | 1500 | 0.029 |
| 10 | 158.6*22.6 | 2014 | 18060 | 900 | 1500 | 0.095 |
| 11 | 132*19 | | | | | |
| 12 | | | | 900 | | 0.02 |
| 13 | 180*28 | 2014 | 28791 | | | 0.3 |
| 14 | 158.5*22.6 | 2015 | 11872 | 900 | 1000 | 0.07 |
| 15 | 140*19.8 | 2009 | 10685 | 220 | 1000 | |
| 16 | 255.1*37.3 | 2012 | 67040 | 600 | 1500 | 0.029 |
| 17 | 180*28 | 2014 | 27821 | | | |
| 18 | | | | | | |
| 19 | 255.1*37.3 | 2013 | 66903 | 1320 | 900 | |
| 20 | | | | | | |

Table 2: Brief information of 20 sampled ships

**Table 23:** Top 32 VOC species mass percentage from 16 container ships

| Compounds | Percentage | Standard deviation | Compounds | Percentage | Standard deviation |
|---|---|---|---|---|---|
| n-Pentane | 0.536278 | 0.629109 | 1-Pentene | 1.325768 | 1.070632 |
| n-Hexane | 1.891022 | 1.481339 | 1-Hexene | 1.457199 | 1.094036 |
| Methylcyclopentane | 1.239405 | 0.691045 | Benzene | 4.190024 | 3.000858 |
| Cyclohexane | 1.241246 | 0.847235 | Toluene | 4.47936 | 1.565238 |
| n-Heptane | 5.313712 | 3.965797 | o-Xylene | 1.379276 | 0.603842 |
| Methylcyclohexane | 6.655522 | 5.044339 | m/p-Xylene | 1.84439 | 0.778013 |
| 3-Methylhexane | 2.397949 | 1.855725 | Ethylbenzene | 1.091218 | 0.472748 |
| 2,3-DimethylPentane | 1.206319 | 1.044379 | m-Ethyltoluene | 1.05578 | 0.652482 |
| 2-Methylhexane | 1.559125 | 1.225618 | 1,2,4-Trimethylbenzene | 1.684083 | 1.045932 |
| n-Octane | 5.798276 | 3.61007 | 1,2,3-Trimethylbenzene | 0.718099 | 0.414425 |
| 3-Methylheptane | 1.918903 | 1.344047 | Acetone | 5.92993 | 5.138703 |
| n-Nonane | 5.844313 | 2.757486 | $CCl_4$ | 1.791729 | 2.854673 |
| 2-Methylheptane | 3.546112 | 2.630706 | $CH_2Cl_2$ | 5.132463 | 10.42498 |
| n-Decane | 5.305601 | 4.532773 | $CH_3Cl$ | 0.681607 | 0.871668 |
| n-Undecane | 5.125647 | 5.741735 | 1,1,2-trichloroethane | 2.610167 | 1.975113 |
| 1-Butene | 1.507865 | 1.176521 | trans-1,2-Dichloroethene | 1.274261 | 1.402993 |

**Table 34:** Element analysis of 3 fuel samples and comparison with previous studies

| | | Celo et al., 2015 | | | This study(Sampled on Jan 14th, 2017) | | |
|---|---|---|---|---|---|---|---|
| Fuel | IFO380 | IFO180 | IFO60 | MDO | MDO(ship 9) | MDO(ship 19) | MDO(unplanned ship) |
| 15℃ density($\text{kg} \cdot m^{-3}$) | 988 | 973.7 | 957.6 | 854.3 | 848.2 | 853.1 | 846.3 |
| w% C | 86.26 | 86.78 | 87.22 | 86.85 | 85.16 | 84.78 | 86.83 |
| w% H | 11.26 | 10.7 | 11.05 | 12.97 | 13.07 | 13.21 | 13.15 |
| w% N | 0.39 | 0.38 | 0.38 | 0.026 | 0.027 | 0.026 | 0.010 |
| w% S | 2.7 | 2.21 | 1.22 | 0.119 | 0.38 | 0.080 | 0.065 |
| mg $\cdot kg^{-1}$ Fe | 31.44 | 17.71 | Not detected | Not detected | 2.7 | <1 | <1 |
| mg $\cdot kg^{-1}$ V | 133.8 | 102.4 | 38.0 | Not detected | Not detected | Not detected | Not detected |
| mg $\cdot kg^{-1}$ Ni | 63.2 | 46.5 | 21.0 | Not detected | Not detected | Not detected | Not detected |
| mg $\cdot kg^{-1}$ Cu | 29.51 | 23.63 | Not detected | Not detected | Not detected | Not detected | Not detected |

IFO: Intermediate Fuel Oil MDO: Marine Diesel Oil

Two of the fuel samples were from ship 9 and ship 19, and the sample marked as unplanned ship indicated

that this ship was not among the 20 ships included in this study.

---

## Author Response (AR3)

**Response to Referee #1**

1. Figure 2: the y-axis is labelled "mass percentage", but the figures are mass fractions, not mass percentages.

**Response:** *Thanks for the careful review. The label of Y-axis in Figure 2 has been revised accordingly.*

**Revision in manuscript:**

[Figure]

2. The corresponding tabulated data in Table 3 have far too many significant figures. I suggest that 2 decimal places at the most would better reflect the uncertainties in the measurements

**Response:** *Thanks for the helpful suggestion. We have revised the decimal places of data in Table.*

**Revision in manuscript:**

**Table3:** Top 32 VOC species mass percentage from 16 container ships

| Compounds | Percentage | Standard deviation | Compounds | Percentage | Standard deviation |
|---|---|---|---|---|---|
| n-Pentane | 0.54 | 0.63 | 1-Pentene | 1.33 | 1.07 |
| n-Hexane | 1.89 | 1.48 | 1-Hexene | 1.46 | 1.10 |
| Methylcyclopentane | 1.24 | 0.70 | Benzene | 4.20 | 3.00 |
| Cyclohexane | 1.24 | 0.85 | Toluene | 4.48 | 1.57 |
| n-Heptane | 5.31 | 3.97 | o-Xylene | 1.38 | 0.60 |
| Methylcyclohexane | 6.66 | 5.04 | m/p-Xylene | 1.84 | 0.78 |
| 3-Methylhexane | 2.40 | 1.86 | Ethylbenzene | 1.10 | 0.47 |
| 2,3-DimethylPentane | 1.21 | 1.04 | m-Ethyltoluene | 1.06 | 0.65 |
| 2-Methylhexane | 1.56 | 1.23 | 1,2,4-Trimethylbenzene | 1.68 | 1.05 |
| n-Octane | 5.80 | 3.61 | 1,2,3-Trimethylbenzene | 0.72 | 0.41 |
| 3-Methylheptane | 1.99 | 1.34 | Acetone | 5.93 | 5.14 |
| n-Nonane | 5.84 | 2.76 | $CCl_4$ | 1.79 | 2.85 |
| 2-Methylheptane | 3.55 | 2.63 | $CH_2Cl_2$ | 5.13 | 10.42 |
| n-Decane | 5.31 | 4.53 | $CH_3Cl$ | 0.68 | 0.87 |
| n-Undecane | 5.13 | 5.74 | 1,1,2-trichloroethane | 2.61 | 1.98 |
| 1-Butene | 1.51 | 1.18 | trans-1,2-Dichloroethene | 1.27 | 1.40 |

**Response to Referee #2**

1. While compounds with higher carbon numbers are more relevant to secondary organic aerosols, how about the effect of lower-carbon compounds on the ozone formation potential? The effects of neglecting lower-carbon compounds on calculated SOA yields and especially OFP should have been evaluated quantitatively.

**Response:** *Yes, we should admit that the absence of lower-carbon compounds in GC-MS quantification has impact on the calculated results of ozone forming potential. According to Gentner's research (Gentner et al., 2012), hydrocarbons with carbon number lower than 6 would not be considered as SOA precursors. Therefore in this revision, we provide supplementary data for low-carbon compounds based on the SIFT-MS quantification to estimate the mass percentage of ethane, ethene, propane, propene and acetylene and their ozone forming potentials, which was not included in previous versions. While such quick online quantification by SIFT-MS might not be as precise as the traditional GC-MS, the results could provide a basic estimation of lighter hydrocarbons.*

**We have made revisions in manuscript regarding detailed analysis in diferent sections:**

**(1) Main text, 2.1.2 VOCs sampling and analysis, Page 6, Line 15- Line 26:**

"In order to complement the quantification of lighter hydrocarbon compounds with 2-3 carbons, selected ion flow tube mass spectrometer (SIFT-MS) was applied, which was recognized as a real-time analytical technique for combustion gases and components in breath(Smith and Spanel, 2010). This part of analysis was done without pretreatments immediately after the gaseous samples from auxiliary engines. VOC samples were transferred from canister to Teflon bag using Agilent headspace syringe with standard volume of 10ml and then diluted with nitrogen. The species that SIFT-MS quantified were not totally consistent with GC-MS. According to the PAMS and TO-15 standard gas applied in GC-MS calibration, 51 species were selected and normalized by mass in SIFT-MS data. Among the 51 species, acetylene, ethane, ethene, propane and propene were the most significant low carbon compounds that were not quantified by GC-MS in this study. Mass proportion of these species was calculated according to the quantification results obtained by SIFT-MS analysis and the impact on ozone forming potentials could be evaluated. "

**(2) Main text, 3.2 SOA and OFP by VOCs from ship exhausts, Page 9 Line 22-Page 10, Line 2:**

"Considering the impact of low carbon hydrocarbons on ozone forming potential, the quantification

results by SIFT-MS were used to evaluate the underestimation caused by the absence of C2-C3 hydrocarbons in GC-MS quantification. A total of 13 auxiliary engine exhaust samples were analyzed by SIFT-MS and the mass fraction of low carbon VOCs as well as their ozone forming potentials were presented as **Fig.4**.

**Fig.4** showed that besides sample 1 and sample 3, the total mass fractions of the low-carbon compounds were below 0.05. Given the fact that SIFT-MS was not able to analyze all 93 VOC species as GC-MS did, the actual fractions could be even lower. When using Maximum Incremental Reactivity scale by Carter's study(Carter, 1994), the calculated OFPs of the low-carbon compounds in sample 1 to sample 13 were between 0.02-0.47 g $O_3$/g VOCs, with an average of 0.14 g $O_3$/g VOCs. Compared to the average value of 2.63 g $O_3$/g VOCs calculated based on the source profile by GC-MS without those hydrocarbons with 2-3 carbon atoms, the underestimation was merely around 5%."

[Figure]

**Figure 4:** Low carbon VOC mass fraction and corresponding ozone forming potentials.

2. Second, while wind direction is recognized as the most significant meteorological conditions, the atmospheric layer stable conditions should also play an important role in the opportunity of the plumes reaching the measurement site. These factors may have large impacts on the analytical results and conclusions in this study.

**Response:** *Your suggestion regarding the atmospheric layer stable conditions is highly appreciated. Several key points that were made related to this issue:*

*(1) We need to clarify that the major part of this study, the emission profile, is directly measured from the flue pipe not gas plume. So these results were not influenced by the meteorological conditions. To avoid misunderstanding, we clarified where the plumes were used and where the flue were used again.*

*(2) Our previous source apportionment only distinguished results before and after DECA policy implementation. Based on the first round referee's comments, we added discussions on wind direction and did see significant impacts on the results. In this revision, we further analyzed layer stable condition's impacts on source apportionment. We have tried our best to collect as many as possible indicators to identify these impacts. The detailed results were shown in this revision.*

*(3) If the ambient atmosphere is not polluted, the plume will have significant impacts on the capture of air pollutants by instruments. However, our test sites and the test season was with heavy polluted days. And the port is located in Tangshan, a typical industrial city with hourly averaged PM2.5 concentrations during our campaign reaching 147 $\mu g \cdot m^{-3}$, and over 85% of the total hours under pollution according to the China air quality standard. The main driving force on ambient pollution is not shipping emissions, but intense local industrial pollution and regional pollution background. In this case, when the atmospheric layer becomes stable, both rapid primary particle accumulation and formation of secondary aerosols occurs concurrently. This phenomenon has been frequently observed in North China and become a persistent unsolved issue in academy (see several previous important papers on this issue, e.g. Science Advances(Cheng et al., 2016), PNAS(Wang et al., 2016), ACP(Zheng et al., 2015a;Zheng et al., 2015b)). In conclusion, it is hard to precisely determine the impacts of stable level of atmospheric layer.*

**According to the key points above, several detailed revisions have been made in the manuscript:**

**(1) Abstract, Page 2, Line 3-Line 6:**

"The impact of atmospheric stability was discussed based on $PM_{2.5}$ and primary pollutants CO concentration. With the background of frequent haze episodes and complex mechanisms of particulate accumulation and secondary formation, the impact of atmospheric stability would be weaken on the sulfate contribution by shipping emissions."

**(2) Main Text: Page 17, Line 11:**

"A new subtitle has been added as **The impact of atmospheric layer stability**"

**(3) Main Text: Page 17, Line 12-Page 18, Line 30:**

"Apart from wind direction, atmospheric layer stability is also a factor that can affect the particle diffusion in exhausted plumes. However, owing to the experimental conditions and the obscure sampling location during the measurements, the actual planetary boundary layer (PBL) height and cloud cover data were not available in this study, which could directly reflect the atmospheric stability. However, atmospheric layer stability is able to be identified based on indirect parameters. According to the study on the correlation between heavy haze episodes and synoptic meteorological conditions in Beijing, China(Zheng et al., 2015b), the polluted periods were associated with lower PBL height, which meant the atmospheric layer was more stable compared with the clean periods. Moreover, according to **Petäjä**'s study(Petäjä et al., 2016), high particulate concentration would in turn decrease the height of boundary layer via feedback mechanisms. Therefore, it can be inferred that the variation of $PM_{2.5}$ and wind speed can indirectly reflect the changes of atmospheric stability. Additionally, the temporal variation of CO concentrations can also reflect the local diffusion conditions as well as layer stability (Xu et al., 2011), since it is a unreactive air pollutant with a long atmospheric lifetime (two months) (Novelli et al., 1998), and mainly from primary emissions.

Based on the discussions above, temporal profiles of $PM_{2.5}$ and CO concentrations as well as the variation of sulfate contribution from ships are presented in **Fig.11**. The CO and $PM_{2.5}$ data is derived from both online monitoring instrument at the sampling site and national monitoring site 20km away. The trends of data from two different sites show excellent correlation with Pearson's r=0.91 and 0.88 respectively for $PM_{2.5}$ and CO (**Fig.11 (a) and (b)**), and this indicates that our measurement results is representative for the ambient atmospheric environment. It can also be observed that the hourly variation of $PM_{2.5}$ concentrations at the sampling site is well agreed with that of CO (r= 0.76), which illustrates that the local air pollution levels was mainly determined by the meteorology conditions, and the trends of the polluted level near the sampling site can be applied in indirectly judging the stable condition of ambient atmosphere. According to the $PM_{2.5}$ concentration threshold suggested by (Zheng et al., 2015b), hourly air pollution levels during the whole measurement period were classified into four categories: clean (hourly $PM_{2.5}<35\mu g \cdot m^{-3}$); slightly polluted ($35\text{-}115\mu g \cdot m^{-3}$); polluted ($115\text{-}350\mu g \cdot m^{-3}$) and heavily polluted ($>350\mu g \cdot m^{-3}$). It was found that the hourly averaged PM2.5 concentrations during our campaign reached 147 $\mu g \cdot m^{-3}$, and over 85% of the total hours were under pollution. Therefore, the atmospheric

layer condition remained stable during most of the time in measurement periods.

While the contribution of sulfate from ships showed an obvious converse trend towards the pollution levels (r=-0.64) (**Fig.11 (c)**). The average contributions were calculated as 32.9%, 27.6%, 19.7% and 15.1% during the clean, slightly polluted, polluted and heavily polluted periods, respectively. That is, with the atmospheric stability went up, the contribution of ships would go down.

The explanation to this phenomenon could be summarized in two aspects. Firstly, during the winter haze periods in China, not only the stable atmospheric condition could increase the total concentration of $PM_{2.5}$, but also the secondary inorganic aerosols, especially sulfates, which were rapidly formed via complicated heterogeneous reactions under high relative humidity (RH)(Wang et al., 2016;Cheng et al., 2016;Zheng et al., 2015a). In our study, the RH in port areas were higher than that in the inland, resulting in the predominance of sulfate formation during the polluted period. The effects of the increased ambient sulfate concentrations on reducing the sulfate relative contributions from ships are much more significant than the increased shipping contributions due to stable atmosphere. Moreover, in addition to the local diffusion conditions, the contribution of shipping emissions to the ambient air quality also highly depended on the real-time density of ships at berth. The number of ships at berth would decrease sharply when heavy haze led to fairly poor visibility. Therefore, less shipping emissions lowered its source contributions during the air pollution period. In summary, the effects of atmospheric layer stability on the sulfate contributions from ships was less important during the haze period compared with that in clean days."

**(4) Conclusion: Page 20, Line 14- Line 17:**

"The impact of atmospheric layer stability reflected by concentration of $PM_{2.5}$ and primary pollutant CO was also discussed and the results showed that with the background of frequent haze episodes and complex mechanisms of particulate accumulation and secondary formation, the impact of atmospheric stability would be weaken on the sulfate contribution by shipping emissions."

**(5) Figure 11 has been added and the numbers of figures have been revised accordingly.**

[Figure]

**Figure 11:** Temporal profiles of PM$_{2.5}$ concentration, CO concentration and sulfate contribution by shipping emissions. (a) PM$_{2.5}$ concentration from two monitoring sites; (b) CO concentration from two monitoring sites; (c) sulfate contribution by shipping emissions.

**Reference**

Carter, W. P. L.: Development of Ozone Reactivity Scales for Volatile Organic Compounds, J.air Waste Man.ass, 44, 881-899, 1994.

Cheng, Y., Zheng, G., Wei, C., Mu, Q., Zheng, B., Wang, Z., Gao, M., Zhang, Q., He, K., and Carmichael, G.: Reactive nitrogen chemistry in aerosol water as a source of sulfate during haze events in China, Science Advances, 2, 2016.

Gentner, D. R., Isaacman, G., Worton, D. R., Chan, A. W. H., Dallmann, T. R., Davis, L., Liu, S., Day, D. A., Russell, L. M., Wilson, K. R., Weber, R., Guha, A., Harley, R. A., and Goldstein, A. H.: Elucidating secondary organic aerosol from diesel and gasoline vehicles through detailed characterization of organic carbon emissions, Proceedings of the National Academy of Sciences of the United States of America, 109, 18318-18323, 10.1073/pnas.1212272109, 2012.

Novelli, P. C., Masarie, K. A., and Lang, P. M.: Distributions and recent changes of carbon monoxide in the lower troposphere, Journal of Geophysical Research Atmospheres, 103, 19015-19033, 1998.

Petäjä, T., Järvi, L., Kerminen, V. M., Ding, A. J., Sun, J. N., Nie, W., Kujansuu, J., Virkkula, A., Yang, X., and Fu, C. B.: Enhanced air pollution via aerosol-boundary layer feedback in China, Sci Rep, 6, 18998, 2016.

Smith, D., and Spanel, P.: Selected ion flow tube mass spectrometry (SIFT-MS) for on-line trace gas analysis, Mass Spectrometry Reviews, 24, 661-700, 2010.

Wang, G., Zhang, R., Gomez, M. E., Yang, L., Zamora, M. L., Hu, M., Lin, Y., Peng, J., Guo, S., and Meng, J.: Persistent sulfate formation from London Fog to Chinese haze, Proceedings of the National Academy Ofences of the United States of America, 48, 13630-13635, 2016.

Xu, W. Y., Zhao, C. S., Ran, L., Deng, Z. Z., Liu, P. F., Ma, N., Lin, W. L., Xu, X. B., Yan, P., He, X., Yu, J., Liang, W. D., and Chen, L. L.: Characteristics of pollutants and their correlation to meteorological conditions at a suburban site in the North China Plain, Atmospheric Chemistry and Physics, 11, 4353-4369, 10.5194/acp-11-4353-2011, 2011.

Zheng, B., Zhang, Q., Zhang, Y., He, K. B., Wang, K., Zheng, G. J., Duan, F. K., Ma, Y. L., and Kimoto, T.: Heterogeneous chemistry: a mechanism missing in current models to explain secondary inorganic aerosol formation during the January 2013 haze episode in North China, Atmospheric Chemistry & Physics, 14, 2031-2049, 2015a.

Zheng, G. J., Duan, F. K., Su, H., Ma, Y. L., Cheng, Y., Zheng, B., Zhang, Q., Huang, T., Kimoto, T., and Chang, D.: Exploring the severe winter haze in Beijing: the impact of synoptic weather, regional transport and heterogeneous reactions, Atmospheric Chemistry & Physics, 15, 2969-2983, 2015b.